# The impact of SF$_6$ sinks on age of air climatologies and trends

Sheena Loeffel[1], Roland Eichinger[2,1,5], Hella Garny[1,2], Thomas Reddmann[3], Frauke Fritsch[1,2], Stefan Versick[4], Gabriele Stiller[3], and Florian Haenel[3]

[1]Deutsches Zentrum für Luft- und Raumfahrt (DLR), Institut für Physik der Atmosphäre, Oberpfaffenhofen, Germany
[2]Ludwig Maximilians Universität, Institut für Meteorologie, Munich, Germany
[3]Karlsruhe Institute of Technology, Institute for Meteorology and Climate Research, Karlsruhe, Germany
[4]Karlsruhe Institute of Technology, Steinbuch Centre for Computing (SCC), Karlsruhe, Germany
[5]Charles University, Department of Atmospheric Physics, Faculty of Mathematics and Physics, Prague, Czech Republic

**Correspondence:** Sheena Loeffel (Sheena.Loeffel@dlr.de)

**Abstract.**

Mean age of air (AoA) is a common diagnostic for the strength of the stratospheric overturning circulation in both climate models and observations. AoA climatologies and its trends over the recent decades of model simulations and proxies derived from observations of long-lived tracers do not agree. Satellite observations show much older air than climate models and while most models compute a clear decrease of AoA over the last decades, a thirty-year timeseries from measurements shows a statistically non-significant positive trend in the NH extratropical middle stratosphere. Measurement-based AoA derivations are often based on observations of the trace gas SF$_6$, a fairly long-lived gas with a near-linear increase of emissions during the recent decades. However, SF$_6$ has chemical sinks in the mesosphere, which are not considered in most model studies. In this study, we explicitly compute the chemical SF$_6$ sinks based on chemical processes in the global chemistry-climate model EMAC. We show that good agreement of stratospheric AoA in EMAC and MIPAS is reached through the inclusion of chemical SF$_6$ sinks, as those lead to a strong increase of the stratospheric AoA and thereby to a better agreement with MIPAS satellite observations. Remaining larger differences in high latitudes are addressed and possible reasons are discussed. Subsequently, we demonstrate that also the AoA trends are strongly influenced by the chemical SF$_6$ sinks. Under consideration of the SF$_6$ sinks, the AoA trends over the recent decades reverse sign from negative to positive. We conduct sensitivity simulations which reveal that this sign reversal results neither from trends of the stratospheric circulation strength, nor from changes in the strength of the SF$_6$ sinks. We illustrate that even a constant SF$_6$ destruction rate causes a positive trend in the derived AoA, since the amount of depleted SF$_6$ scales with the increasing SF$_6$ abundance itself. In our simulations, this effect overcompensates the impact of the accelerating stratospheric circulation which naturally decreases AoA. Although various sources of uncertainties cannot be quantified in detail in this study, our results suggest that the inclusion of SF$_6$ depletion in models has the potential to reconcile the AoA trends of models and observations. We conclude the study with a first approach towards a correction to account for SF$_6$ loss and deduce that a linear correction might be applicable to values of AoA of up to 4 years.

# 1 Introduction

The Brewer-Dobson circulation (BDC) describes the stratospheric transport circulation, consisting of the mean overturning circulation of air ascending in the tropical pipe, moving poleward and descending in the extratropics (Brewer, 1949; Dobson and Massey, 1956), as well as isentropic mixing. A good measure to diagnose this transport circulation is the age of stratospheric air (AoA), which is defined as the mean transport time of an air parcel from its entry into the stratosphere (or from the surface) to any point therein (Hall and Plumb, 1994; Waugh and Hall, 2002). In general circulation models (GCMs), AoA is commonly represented by an inert tracer with a strictly linear temporally increasing surface mixing ratio and is calculated as the corresponding time lag between the local mixing ratio and the mixing ratio at a reference point (Hall and Plumb, 1994). The same method can be applied to real long-lived tracers with a linear trend in tropospheric concentration and AoA has been derived, for example, from balloon-borne in situ measurements of sulphur hexafluoride ($SF_6$) (Andrews et al., 2001; Engel et al., 2009, 2017). This trace gas is particularly suitable for these studies as it is stable in the troposphere and stratosphere and its tropospheric concentrations increased nearly linearly over the recent decades. Together with observations of other trace gases, these measurements form a long-term dataset of observationally based AoA restricted to the Northern Hemisphere mid-latitudes that covers more than 40 years. A near global dataset of AoA covering ten years from 2002-2012 was derived in Stiller et al. (2012), Haenel et al. (2015) and Stiller et al. (2017), who retrieved $SF_6$ distributions from MIPAS (Michelson Interferometer for Passive Atmospheric Sounding) satellite observations, but these cover a much shorter time period.

Observations and model simulations of AoA often disagree. AoA derived from observations is mainly older than simulated AoA (see e.g. SPARC, 2010; Dietmüller et al., 2018) and the AoA trend over recent decades even differs in sign between observations and models. While most climate models show a clear decrease of AoA over time (see e.g. Butchart and Scaife, 2001; Garcia et al., 2011; Eichinger et al., 2019), consistent with the simulated acceleration of the BDC in the course of climate change (see e.g. Garcia and Randel, 2008), the time series of the observations presented in the studies by Engel et al. (2009), Ray et al. (2014) and Engel et al. (2017) show a (statistically non-significant) positive trend (note that Ray et al. (2014) also shows negative balloon AoA trends in the lower extratropical stratosphere). This discrepancy has been addressed in numerous studies: Garcia et al. (2011) showed that due to the concave growth rate of tropospheric $SF_6$ concentrations, the AoA trends derived from an $SF_6$ tracer are smaller than the trends derived from a synthetic, linearly growing AoA tracer (also after accounting for the nonlinear growth rates of $SF_6$). They noted that the sparse sampling of in situ observations can be the reason for the above mentioned trend discrepancies. Birner and Bönisch (2011) as well as Bönisch et al. (2011) argued that differences in the changes between the deep and the shallow BDC branch can possibly explain these. Ploeger et al. (2015) showed that the residual circulation transit time cannot explain the AoA trends, and that the integrated effect of mixing (which is coupled to residual circulation changes, see Garny et al., 2014) is crucial. Moreover, Stiller et al. (2017) could explain a hemispheric asymmetry by a shift of subtropical transport barriers. In a study based on a chemistry transport model, Kouznetsov et al. (2020) showed that changes in $SF_6$-derived apparent AoA over one decade are highly influenced by the $SF_6$ sink, and can even turn positive. However, a comprehensive understanding of what contribution the individual effects have on the AoA trend depending on altitude and latitude is still missing.

SF$_6$ sinks lead to older apparent AoA (see, e.g. Waugh and Hall (2002) and Kouznetsov et al. (2020)), as well as shorter lifetimes. Leedham Elvidge et al. (2018) evaluated AoA from several tracers including SF$_6$ and found clear differences between these, which indicated a shorter SF$_6$ lifetime than previously assumed. The strongest chemical SF$_6$ removal reactions take place in the mesosphere, the most important removal processes are electron attachment and UV-photolysis, but these processes have not yet been precisely quantified. Ravishankara et al. (1993) estimated an SF$_6$ lifetime of 3200 years and Reddmann et al. (2001) found a lifetime between 400 and 10000 years, depending on the assumed loss reactions and electron density. A more recent model study by Kovács et al. (2017), who used the Whole Atmosphere Community Climate Model (WACCM) to determine the atmospheric lifetime of SF$_6$, reported a mean SF$_6$ lifetime of 1278 years and Ray et al. (2017) provided a range between 580 and 1400 years based on in situ measurements in the stratospheric polar vortex. The most recent model study of Kouznetsov et al. (2020), who performed simulations of tracer transport with a chemical transport model, shows an SF$_6$ lifetime ranging between 600 and 2900 years. Due to these uncertainties and the complex computation of the chemical reactions, most model studies do not consider any SF$_6$ sinks for the calculation of AoA from SF$_6$ mixing ratios. This can explain why most climate models generally show younger stratospheric air than observations, in particular within the polar vortices (e.g. Haenel et al., 2015; Ray et al., 2017).

In the present study, we apply the chemistry climate model EMAC (ECHAM MESSy Atmospheric Chemistry, Jöckel et al., 2010; Jöckel et al., 2016) with the aim to understand the effects of SF$_6$ sinks on tracer-derived AoA and its long-term trends. Specifically, we calculate for the first time the effect of the sinks on the long-term trend of SF$_6$-derived AoA, and quantify how this effect is modulated by circulation changes (recent climate change), specified model dynamics, or by changes in the abundance of relevant species for SF$_6$ chemistry. Furthermore, we analyse the contribution of the SF$_6$ sinks themselves on the long-term trend of SF$_6$-based AoA. As an outlook, we thereupon provide first thoughts on how to apply an AoA correction to observations taking SF$_6$ sinks into account. The chemistry climate model uses the second version of the Modular Earth Submodel System (MESSy2) to link multi-institutional computer codes. In our simulations, we employed the MESSy submodul "SF6" which explicitly calculates SF$_6$ sinks based on physical processes (based on Reddmann et al., 2001), rather than on crude parameterisations. We apply a correction for the non-linear growth of SF$_6$ in the calculation of AoA, based on Fritsch et al. (2020), which allows for the quantification of the effect of SF$_6$ sinks on SF$_6$-based AoA in isolation. In Sect. 2 we describe the EMAC model and the SF6 submodel as well as the observational data we use for comparison. Sect. 3 contains a comparison of the EMAC climatologies with MIPAS data, a comparison of the EMAC trends with MIPAS and balloon borne measurements and an analysis of the results of two sensitivity simulations. The model results are discussed in the following using theoretical considerations of the effects of sinks on AoA trends (Sect. 4), including first thoughts on possible correction methods for the sinks, that are highly desirable for the use of observational data. In Sect. 5, we discuss the results and provide some concluding remarks.

## 2 Atmospheric Model

### 2.1 EMAC model

For this study, we use the EMAC (ECHAM MESSy Atmospheric Chemistry, v2.54.0, Jöckel et al. (2010); Jöckel et al. (2016)) model, a numerical Chemistry and Climate Model (CCM) system. It contains the General Circulation Model (GCM) ECHAM5 (ECMWF Hamburg, Roeckner et al. (2003)), with its spectral dynamical core, as well as the MESSy (Modular Earth Submodel System, Jöckel et al. (2005); Jöckel et al. (2010)) submodel coupling interface. The latter is a modular interface structure for the standardised control of process-based modules (submodels) and their interconnections. We apply the model in a T42 horizontal ($\sim 2.8°$x$2.8°$) resolution with 90 layers in the vertical and explicitly resolved middle atmosphere dynamics (T42L90MA). In this setup, the uppermost model layer is centred at 0.01 hPa and the vertical resolution in the upper troposphere lower stratosphere region (UTLS) is 500-600 m. In the standard reference setup, we use the basic EMAC modules for dynamics, radiation, clouds, and diagnostics (AEROPT, CLOUD, CLOUDOPT, CVTRANS, E5VDIFF, GWAVE, ORBIT, OROGW, PTRAC, QBO, RAD, SURFACE, TNUDGE, TROPOP, VAXTRA, refer to Jöckel et al., 2005; Jöckel et al., 2010, for details on these submodels). Additionally we included the new submodel SF6.

### 2.2 Submodel SF6

The submodel SF6 is used to calculate the lifetime of $SF_6$ by explicitly accounting for the sinks of $SF_6$ in the mesosphere. The submodel is operationally available for all users in MESSy from version 2.54.0 onward. The calculation method for this is based on the reaction scheme of Reddmann et al. (2001). The most important reaction involved in the chemical degradation of $SF_6$, namely electron attachment, is included in the $SF_6$ submodel. The configuration of the submodel allows for a simple exponential profile for the electron field and a more complex field based on Brasseur and Solomon (1986), whereas in the present study we use the latter option. It depends on altitude, latitude, solar zenith angle, air density and day of year. In contrast to Reddmann et al. (2001), UV-photolysis of $SF_6$ is not included in the submodel. The loss rate by photolysis is several orders of magnitude weaker than that of electron attachment up to altitudes of about 100 km (see e.g. Fig. 9 in Totterdill et al., 2015) and is therefore not relevant for the focus of our study. Further reactions considered are photodetachment of $SF_6^-$ (Datskos et al., 1995), destruction of $SF_6^-$ by atomic hydrogen, hydrogen chloride and ozone (Huey et al., 1995), stabilisation of excited $SF_6^-$ by collisions and autodetachment of $SF_6^-$. An overview is provided in Table 1. Reddmann et al. (2001) used climatological profiles for the forementioned gases, while in our submodel channel objects (see Jöckel et al., 2016) are used. Such channel objects can be calculated in other submodels (e.g. interactive chemistry), prescribed as external time series (in this study) or just be simple climatologies. The autodetachment rate can be chosen in the namelist and was set to $10^6 s^{-1}$ (see Reddmann et al., 2001). For a general overview of the various reactions see Fig. S1 in the Supplementary Information.

**Table 1.** Chemical reactions of $SF_6$. The labelling of the various reactions mirrors the style used by Reddmann et al. (2001). Reactions labelled with † are included in the SF6 submodel.

| Reaction No. | Reaction | Description |
|---|---|---|
| (R1) | $SF_6 + h\nu \longrightarrow SF_5 + F$ | destructive UV-photolysis |
| (R2) | $^\dagger SF_6 + e^- \longrightarrow (SF_6^-)^*$ <br> $SF_6 + O^+ \longrightarrow SF_5^+ + OF$ <br> $SF_6 + N_2^+ \longrightarrow SF_5^+ + NF$ <br> $SF_6 + O_2^- \longrightarrow SF_6^- + O_2$ | destructive Electron Attachment and Secondary Reactions |
| (R3) | $^\dagger SF_6^- + h\nu \longrightarrow SF_6 + e^-$ | photodetachment |
| (R4) | $^\dagger SF_6^- + H \longrightarrow SF_5^+ + HF$ | destructive |
| (R5) | $^\dagger (SF_6^-)^* + M \longrightarrow SF_6^-$ | stabilisation against autodetachment |
| (R6) | $^\dagger (SF_6^-)^* \longrightarrow SF_6 + e^-$ | autodetachment |
| (R7) | $^\dagger SF_6^- + HCl \longrightarrow$ products <br> $SF_6^- + HNO_3 \longrightarrow$ products <br> $SF_6^- + SO_2 \longrightarrow$ products | destructive |
| (R8) | $^\dagger SF_6^- + O_3 \longrightarrow SF_6 + O_3^-$ <br> $SF_6^- + O \longrightarrow SF_6 + O^-$ <br> $SF_6^- + NO_2 \longrightarrow SF_6 + NO_2^-$ | recovery reaction |

## 2.3 Simulation setup

The simulations performed in this study include a more comprehensive approach for the calculation of the $SF_6$ sinks. We use a climate chemistry model (as opposed to studies based on chemistry transport models as e.g. in Kouznetsov et al., 2020) and use a more comprehensive SF6 submodel than previous chemistry climate model studies (see, e.g. Marsh et al., 2013, for the Whole Atmosphere Community Climate Model (WACCM)). Other than the SF6 submodel, no interactive chemistry is activated in the simulations for this study. The reactant species involved in the $SF_6$ chemistry (HCl, H, $N_2$, $O_2$, O($^3$P) and $O_3$) and the radiatively active gases ($CO_2$, $CH_4$, $N_2O$, $O_3$) are transiently prescribed from the ESCiMo RC1-base-07 simulation (see Jöckel et al., 2016) as monthly and zonal means. Moreover, we prescribe the Hadley Centre Sea Ice and Sea Surface Temperature dataset (HADISST), the CCMI-1 volcanic aerosol dataset (for its effect on infrared radiative heating, see Arfeuille et al., 2013; Morgenstern et al., 2017) and QBO nudging (see Jöckel et al., 2016). To compute the photodetachment rate of $SF_6^-$, we follow Reddmann et al. (2001) using the extraterrestial solar photon flux with no attenuation of the UV-photon flux, as provided by WMO (1986). Our simulations range from 1950 to 2011, whereas at least the first ten years have to be considered as a spin-up period. The projection simulation runs from 1950 to 2100 with the $SF_6$ reactant species and GHGs prescribed from the ESCiMo RC2-base-04 simulation (see Jöckel et al., 2016) as monthly and zonal means. In addition to the reference (REF) simulation, we performed two sensitivity simulations and one specified dynamics simulation. The two sensitivity simulations

are as follows: The CSS (constant reaction partners for $SF_6$ sinks) sensitivity simulation differs from the REF simulation only by constant mixing ratios of the reactant species (see above) that influence the $SF_6$ sinks. For that purpose, we kept the mixing ratios at the level of the start of the simulation (year 1950 on repeat). With this simulation we aim to address the influence of the reactant species involved in the $SF_6$ sink reactions. The second sensitivity simulation, referred to as TS2000, is not a transient simulation as is the REF simulation, but instead a time slice simulation with climate conditions (GHGs, SSTs and SICs) of the year 2000 (climatology of the period 1995-2004). Furthermore, the reactant species for the SF6 sinks have been averaged over the period 1995-2004 for the TS2000 simulation. This will allow us to investigate the effects of the $SF_6$ sinks under a constant climate. In the specified dynamics (SD) simulation, we apply newtonian relaxation ('nudging') towards ERA-INTERIM (Dee et al., 2011) reanalysis data of potential vorticity, divergence, temperature and the logarithmic surface pressure up to 1 hPa. This assures that the meteorological situation largely resembles the ERA-Interim data. The flexible structure of the Modular Earth Submodel System (MESSy, Jöckel et al., 2005) allows us to use the same executable for all simulations, with the differences between them realised through changes in aforementioned namelist settings (see Jöckel et al., 2005). A summary of the simulations used in this study can be found in Table 2.

## 2.4 Satellite and in situ data

The MIPAS (Michelson Interferometer for Passive Atmospheric Sounding) instrument on Envisat (Environmental Satellite) allowed for the retrieval of $SF_6$ by measuring the thermal emission in the mid-infrared, while orbiting the Earth sun-synchronously 14 times a day. This high-resolution Fourier transform spectrometer measured at the atmospheric limb and provided data for $SF_6$ retrievals in full spectral resolution from 2002 to 2004 and in reduced resolution from 2005 to 2012 between 6 and 40 km of altitude (Stiller et al., 2012; Haenel et al., 2015). In this study, a newer version of the MIPAS dataset existing as of 2019 will be shown (see e.g. Stiller, 2021, for AoA), whereby new $SF_6$ absorption cross-sections have been used for the $SF_6$ retrieval (Stiller et al., 2020; Harrison, 2020). Except for the newer absorption cross-sections for $SF_6$ and the accounting for a trichlorofluoromethane (CFC11) band in the vicinity of the $SF_6$ signature, the $SF_6$ retrieval and conversion into AoA was done according to the description by Haenel et al. (2015). In particular, the Level-1b data version is still V5.

Engel et al. (2009) collected available air samples of $SF_6$ and $CO_2$ from balloon-borne cryogenic whole air-samplers flown during 27 balloon flights, with data up to 43 km, and re-analysed these samples in a self-consistent manner. The derived $SF_6$ data cover the years 1975 to 2005 (with a gap between 1985 and 1994) and covers the mid-latitudes between 32°N and 51°N. Since AoA profiles from midlatitudes above approximately 25 km or 30 hPa are constant over altitude, a mean mid-latitude middle stratospheric AoA value from each profile was determined by averaging the vertical profile between 30 hPa and the top balloon flight height. With this procedure, a time series of mid-latitude middle stratospheric AoA values could be determined back to 1975. It is important to note that part of the Engel et al. (2009) AoA time series is derived from $CO_2$ measurements. In this paper, only the AoA data points derived from $SF_6$ are used. Engel et al. (2017) extended the initial dataset from Engel et al. (2009) to 2016, however the AoA here is derived from $CO_2$ measurements and thus also excluded.

**Table 2.** Overview of simulations undertaken in this study

| Simulation | Details |
|---|---|
| Reference (REF) | Transient<br>1950 - 2011<br>Greenhouse gases ($CO_2$, $CH_4$, $N_2O$, $O_3$) transiently<br>prescribed from ESCiMo RC1-base-07-simulation<br>(Jöckel et al., 2016) as monthly and zonal means |
| Specified Dynamics (SD) | Transient<br>1980 - 2011<br>Newtonian relaxation of dynamics towards<br>ERA-Interim reanalysis data (Dee et al., 2011)<br>up to 1hPa |
| **Sensitivity Simulations** | |
| Constant reaction partners<br>for $SF_6$ Sinks (CSS) | Transient<br>1950 - 2011<br>Same conditions as the REF simulation, but for<br>the $SF_6$ reactant species the year 1950<br>concentrations are repeated throughout the model run |
| Time slice (TS2000) | Time slice<br>1950 - 2011<br>Climate conditions (GHGs, SSTs, SICs) of year 2000<br>Climatology taken as 1995 – 2004<br>$SF_6$ sinks reactant species averaged over 1995 – 2004 |
| **Projection Simulation** | |
| Climate Projection (PRO) | Transient<br>1950 - 2099<br>Greenhouse gases ($CO_2$, $CH_4$, $N_2O$, $O_3$) transiently<br>prescribed from ESCiMo RC2-base-04-simulation<br>(Jöckel et al., 2016) as monthly and zonal means |

## 2.5 Analysis method

The basic concepts for the calculation of mean AoA are introduced in Hall and Plumb (1994). In the case of a tracer with a
linear increasing lower boundary condition, AoA can be determined by the time lag between the mixing ratio at a given point in
the atmosphere and the same mixing ratio of the reference time series. As for any realistic tracer, $SF_6$ does not exhibit perfectly
linear growth, for which adjustments in the AoA calculation are needed. This study follows the calculation method employed
by e.g. Engel et al. (2009), which was introduced in Volk et al. (1997). The calculation uses a polynomial fit to the reference
time series to approximate mean AoA. However, in our study we modified the parameters compared to those used in Engel
et al. (2009) to ensure that (passive) $SF_6$-based AoA agrees with the ideal AoA derived from the linear tracer, following Fritsch
et al. (2020). Specifically, in our calculations we used a ratio of moments of 1.0 years and a fraction of input of 95 %. Further,
we used the $SF_6$ mixing ratio averaged over 20°S-20°N at the ground as the reference time series. Due to the availability of
data, this reference region is also used in Engel et al. (2009) with balloon-borne observations.

For the derivation of AoA from MIPAS $SF_6$ observations, Stiller et al. (2008, 2012) and Haenel et al. (2015) used a slightly
smoothed version of the global mean of $SF_6$ surface measurements as the reference time series instead of $SF_6$ at the strato-
spheric entry point, which is not available from observations (see e.g. Dlugokencky, 2005). The non-linearity of the reference
curve was considered by its convolution with an idealized age spectrum parameterized as a function of the mean age within an
iterative approach. For more details, see Stiller et al. (2012) and Haenel et al. (2015).

In our simulations the AoA calculations are applied to a total of four tracers, which can be organised into two groups. The
first assumes a strict linear growth of $SF_6$, producing a linear reference curve, while the second considers a realistic growth of
$SF_6$ based on observed emissions, creating a non-linear $SF_6$ reference curve. Technically, in our simulations these "emissions"
are realised via lower boundary conditions, which are based on surface observations, as in Jöckel et al. (2016). As previously
mentioned, $SF_6$ undergoes chemical degradation predominantly in the mesosphere. Consequently, the absence or presence of
mesospheric sinks is additionally considered, resulting in a total of four tracers tr(WS, $SF_6$), tr(NS, $SF_6$), tr(WS, lin), and tr(NS,
lin). The labelling of these depends on the chemistry involved (with sinks: "WS", without (no) sinks: "NS") and the growth
assumed (linear: "lin", non-linear: "$SF_6$"), and follows the pattern $tr$(CHEMISTRY, GROWTH). When referring to simulations
with a specific tracer, the labelling will follow the notation SIMULATION(CHEMISTRY, GROWTH), and similarly we use the
following notation for AoA inferred from the tracer used in a simulation: AOA(CHEMISTRY, GROWTH)$_{SIM}$. For example:
AoA from the reference simulation based on the tracer with mesospheric sinks and non-linear ($SF_6$ emission based) increase is
referred to as AoA(WS, $SF_6$)$_{REF}$. AoA inferred from the linear tracer without mesospheric sinks in the reference simulation
is denoted as AoA(NS, lin)$_{REF}$.

## 3 Results

### 3.1 SF$_6$ mixing ratios

In order to evaluate the SF$_6$ mixing ratios simulated of the EMAC model, we first analyse the four tracers in the SD simulation
in comparison to observational data. This study does not perform a detailed comparison of SF$_6$ profiles, as the major aim is
not an in-depth evaluation of the SF6 submodel, but rather a quantification of the potential effects of the SF$_6$ sinks on AoA and
its long-term trends. However, to ensure that SF$_6$ values in the EMAC model are within the range of observational estimates,
we perform selected comparisons to data from Ray et al. (2017) and MIPAS SF$_6$ (Stiller et al., 2020, paper in preparation).
Fig.1a depicts the modelled SF$_6$ vertical profile climatologies in comparison with MIPAS SF$_6$. Fig.1b shows the modelled SF$_6$
vertical profiles and balloon-borne measurements of SF$_6$ (Ray et al., 2017) on a particular day. The former comparison is for
zonal mean SF$_6$ averaged over 30°N-50°N and 2007-2010. These years have been chosen as the dataset is complete in this
period. The error bars represent the standard deviation of the zonal mean ensemble, which consists of the measurement noise
error of MIPAS, further random error sources from the retrieval (e.g. temperature uncertainties), the natural variability over the
longitudes of the latitude band, and the four years of averaging (2007 - 2010).

Fig.1b shows modelled SF$_6$ mixing ratios of the day of the balloon flight. The balloon was launched on March 5th, 2000 at
67°N in Kiruna, Sweden. To ensure that the SF$_6$ profile is based on air masses from within the vortex, modelled SF$_6$ values
are averaged over 0°E-100°E and 65°N-80°N, which corresponds to the area of the vortex core for the given day. The standard
deviation of SF$_6$ for 0°E-100°E averaged over the respective latitude range is shown as error bars.

The tracers with non-linear growth in the SD simulation show smaller tropospheric SF$_6$ mixing ratios than the linear tracers
(Fig.1a and b). This can be explained by the two different growth scenarios of SF$_6$ and the prescribed lower boundary conditions
(see Fig. S2 in Supplementary Information). The sinks do not have a considerable effect in the troposphere, hence the effect of
the SF$_6$ sinks becomes only noticeable higher up. Furthermore, the effect of the SF$_6$ depletion becomes increasingly evident
with altitude. This is portrayed in the growing differences with altitude between $tr$(WS, lin) and $tr$(NS, lin) and the non-linear
equivalent. The differences particularly increase for the tracers with linear emissions, as these exhibit higher SF$_6$ mixing ratios
and hence experience greater SF$_6$ depletion than those with non-linear boundary conditions. Due to the small turnaround times
for air in the middle atmosphere, the tracers without sinks exhibit a very low decrease of the SF$_6$ mixing ratios with altitude.

Fig.1a shows that the EMAC simulated non-linear SF$_6$ is within the observed range of MIPAS SF$_6$. Below 30 km, MIPAS
SF$_6$ mixing ratios are smaller, with a near-constant offset of approximately 0.5 pmol/mol up to 20 km. Above 30 km, MIPAS
SF$_6$ shows larger mixing ratios than EMAC. This means that EMAC SF$_6$ (SD(WS, SF$_6$)) shows a larger decrease with altitude
than MIPAS SF$_6$, suggesting that the sinks in EMAC are too strong. Another explanation could be too strong vertical mixing in
EMAC. However, the EMAC SF$_6$ lies within the MIPAS uncertainty range throughout the atmosphere. The standard deviation
increase with height in MIPAS SF$_6$ can be attributed to the decrease of the SF$_6$ signal with height, which leads to an increase in
noise-error of SF$_6$. Additionally, the natural variability of SF$_6$ itself, as well as the evolution of SF$_6$ over time, contribute to the
increasing standard deviation in the MIPAS SF$_6$ profile. The increase in standard deviation with height can also be seen in the

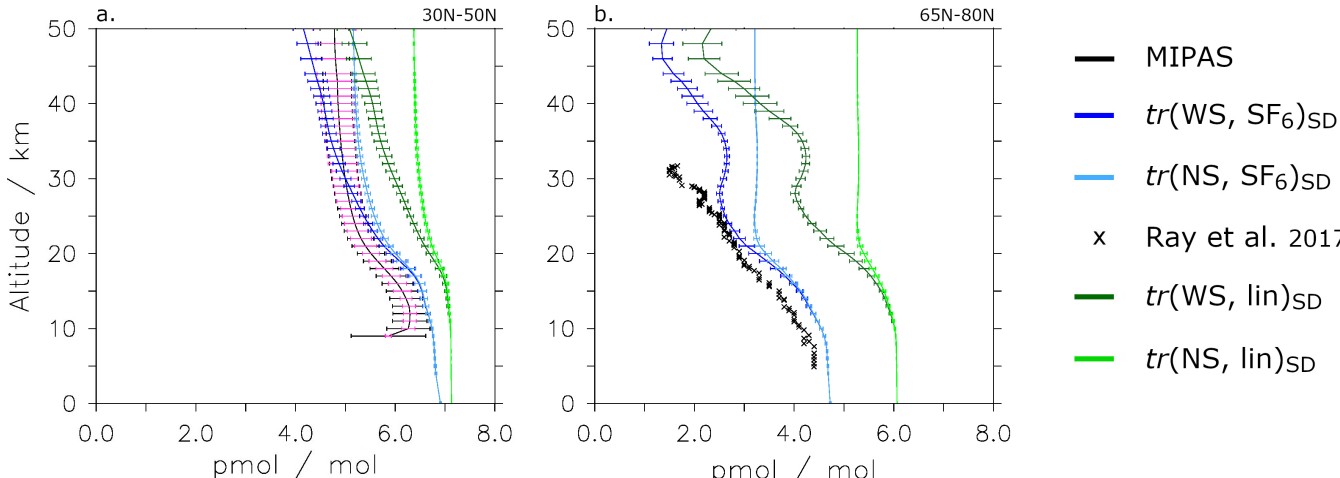

**Figure 1.** Vertical SF$_6$ profiles for the four tracers from the SD simulation averaged over (a) 30°N-50°N (zonally averaged) for 2007-2010 and (b) 65°N-80°N, 0°E-100°E for 5th March 2000. Horizontal lines show the SF$_6$ spread over the selected longitudes. Dark blue: non-linear tracer with sinks $tr$(WS, SF$_6$); light blue: non-linear tracer without sinks $tr$(NS, SF$_6$); light green: idealised tracer $tr$(NS, lin); dark green: linear tracer with sinks $tr$(WS, lin); In panel a), the SF$_6$ mixing ratios obtained from MIPAS (Stiller et al., 2020, and paper in preparation) are shown in black. Black error bars depict the standard deviation of MIPAS SF$_6$, pink error bars the systematic error of MIPAS. The systematic error comprises of errors in the spectroscopic data and uncertainty in the instrumental lineshape, which results in a systematic error of 2% for the lower (10 km) and 11% for the upper (60 km) stratosphere. See Stiller et al. (2020) for details. Black crosses in panel b) represent the balloon-borne measurements (Ray et al., 2017) taken on March 5th, 2000 at Kiruna, Sweden (67°N).

EMAC SF$_6$ profiles, particularly in the tracers $tr$(WS, SF$_6$) and $tr$(WS, lin). However, it is by far not as large as in the MIPAS data because the simulations have no measurement error and possibly show a smaller natural variability than the observations.

The balloon flight SF$_6$ profile (Ray et al., 2014) in Fig.1b largely resembles the profile of the realistically modelled tracer $tr$(WS, SF$_6$). Below 25 km, the modelled SF$_6$ profile shows a constant high bias of around 0.3 pmol/mol, presumably due to the lower boundary conditions used. Larger discrepancies can be seen between 25-35 km altitude, with higher mixing ratios of

230 the modelled SF$_6$. As the data presented in Fig.1b are only for a specific day and region, the particular meteorological situation and location of the balloon can be crucial for the comparison.

### 3.2 SF$_6$ lifetimes

The atmospheric lifetime of SF$_6$ can be used as an indicator for the accuracy of the SF$_6$ degradation scheme. We calculate the lifetime following Equation 1 in Section 3 of Reddmann et al. (2001), namely:

$$\frac{d[SF_6]}{dt} = -[k_1 + k_2(1 - \epsilon\eta)][SF_6] \tag{1}$$

It is based on the reaction rates ($k_i$) of the chemical reactions ($R_i$) marked in Table 1, the branching fraction $\epsilon$ (taken as 0.999, see Reddmann et al. (2001)), and the efficiency of the $SF_6$-recovery reactions ($\eta$), where $\eta$ is calculated as:

$$\eta = \frac{k_5(k_3 + k_8) + k_6(k_3 + k_4 + k_7 + k_8)}{(k_5 + k_6)(k_3 + k_4 + k_7 + k_8)} \tag{2}$$

Only the realistic tracer $tr(\text{WS}, SF_6)$ is considered. The reference simulation yields an average lifetime of 2101 years. Red-

240 dmann et al. (2001) carried out sensitivity experiments with this scheme, using various options for the chemical mechanisms. In this way, the lifetime could be varied between 400 and 10000 years. Our value is below the value 3200 years calculated by Ravishankara et al. (1993), but above the numbers 1278 years and 850 years of the more recent studies by Kovács et al. (2017) and Ray et al. (2017), respectively. In another new modelling study, Kouznetsov et al. (2020) presented a range of 600-2900 years. Our value of around 2100 years is therefore within, but rather at the upper range of the uncertainties. In contrast

to the comparison of the model results with $SF_6$ observations shown in the previous section, our lifetime value points towards rather weak $SF_6$ sinks in our scheme. To assess the variability of the atmospheric $SF_6$ lifetime, we show in Fig.2 the timeseries of the $SF_6$ lifetimes for the four simulations that were described in Sect. 2. The lifetime of the REF simulation lies at about

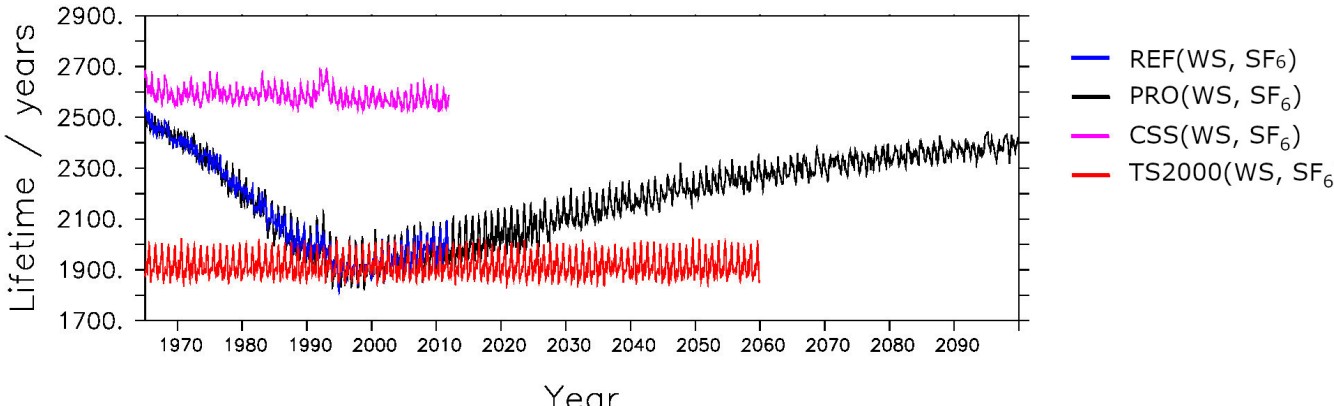

**Figure 2.** Global stratospheric and mesospheric lifetimes of $SF_6$ calculated from the tracer with realistic lower boundary conditions and $SF_6$ sinks ($tr(\text{WS}, SF_6)$). Blue: reference simulation (REF); black: projection simulation (PRO); pink: constant reactant species simulation (CSS); red: time slice 2000 simulation (TS2000).

2500 years in 1965 and decreases by approximately 25% to 1900 years in 2011. The lifetime of the projection simulation behaves similarily, and increases to about 2400 years by the year 2100. This shape of the lifetime resembles that of projected

$O_3$ (see e.g. Eyring et al., 2007), which is reflected in the ESCiMo simulations (Jöckel et al., 2016) from which the $O_3$ and the other $SF_6$-reactant species are prescribed here. However, our $SF_6$ degradation scheme includes a number of simplifications, which can modify the lifetimes. For example, a constant profile is used to prescribe the sinks through NO. This can simplify the long term variability of the $SF_6$ lifetimes by making it overly dependent on the species that are transiently prescribed in the simulations. Apart from seasonal variations, the lifetimes of the CSS and the TS2000 simulations are fairly constant, their life-

time values lie around 2600 and 1900 years, respectively. The fact that the lifetimes of the CSS simulations are fairly constant

implies that the long-term trends of the $SF_6$ lifetimes can mostly be attributed to the abundance of the species involved in the $SF_6$ degradation. Variations of stratospheric temperatures or the circulation strength seem to only play a minor role.

## 3.3 Age of air climatologies

AoA climatologies averaged over 2007 to 2010 from the reference simulation are shown in Fig. 3a-d (REF(NS, lin), REF(NS, $SF_6$), REF(WS, lin), and REF(WS, $SF_6$), from the specified dynamics simulation (SD(WS, $SF_6$)) in Fig. 3e and from MIPAS $SF_6$ observations in Fig. 3f (Stiller, 2021). The years 2007 to 2010 were chosen as these are the only years with complete MIPAS data.

In all cases, AoA increases with increasing altitude and latitude. The cases considering sinks show older apparent AoA than those without, especially with increasing altitude and latitude. This apparent ageing can be explained by the fact that the inclusion of mesospheric sinks results in smaller $SF_6$ mixing ratios. The reduced mixing ratios lead to seemingly older AoA as the corresponding reference value lies further in the past. Downwelling within the polar vortices transports old air from the mesosphere to the stratosphere. With the breakdown of the polar vortex at the end of the winter season, the old air is then mixed into lower latitudes. The relative effect of the sinks on AoA derived from the $SF_6$ tracers with non-linear growth can be seen in Figs. 3g) and 3h). Mean AoA values derived from $SF_6$ in the early period (1970-1980) are moderately affected by the sinks, with a difference of around 20-25% in the polar middle stratosphere and above (i.e., for mean AoA values above 5.5 years). Differences are small (less than 10%) for mean AoA values below 4 years. However, as will be discussed (see Section 3.5), the effect of the $SF_6$ sinks increases over time and for the later period (2000-2010), mean AoA derived from $SF_6$ is affected by the sinks considerably. Differences greater than 20% can be seen in Fig. 3h) for AoA above about 3 years, and in the extratropical lower stratosphere, differences are larger than 10% even for mean AoA values of 2 years and above.

The patterns of modelled AoA from the linear tracers are similar to those of the $SF_6$ emissions-based tracers (compare Fig. 3a with Fig. 3b, and Fig. 3c with Fig. 3d). However, the similarities are weaker in the polar regions, especially when the $SF_6$ sinks are considered (Fig. 3c and Fig. 3d). In these regions, REF(WS, lin) reaches AoA values spanning 10–15+ years, and REF(WS, $SF_6$) 9–14 years. This difference can be attributed to the greater initial growth of the tracer with linear emissions than that with non-linear emissions (Fig. S2 in Supplementary Information). This leads to enhanced $SF_6$ mixing ratios in the REF(WS, lin) case and in turn strengthens the influence of $SF_6$ sinks on AoA. This is particularly relevant in the winter months when $SF_6$-depleted mesospheric air is transported downward into the polar stratosphere.

AoA derived from $SF_6$ emissions including chemical $SF_6$ sinks (REF(WS, $SF_6$), Fig. 3d) agrees best with MIPAS AoA (Fig. 3f). Overall good agreement between EMAC and MIPAS AoA is found in the tropics, but there is a large high bias in the high latitudes in EMAC: between 40°N-50°N, modelled AoA is up to 2 years older than MIPAS AoA in the stratosphere, and up to 3 years older in the polar upper stratosphere (see Fig. S5 in Supplementary Information). In comparison to the MIPAS observations, the $SF_6$ sinks therefore seem to be too strong in the model, as already mentioned above. However, in comparison with the previously published MIPAS data (Stiller et al., 2012; Haenel et al., 2015), the EMAC AoA was actually too young, i.e. the MIPAS AoA was much older in the previous versions in the polar regions. The spectroscopic data used for the $SF_6$ retrieval in MIPAS cause a rather large bias (that has now been corrected by improved spectroscopy). The new spectroscopic data lead

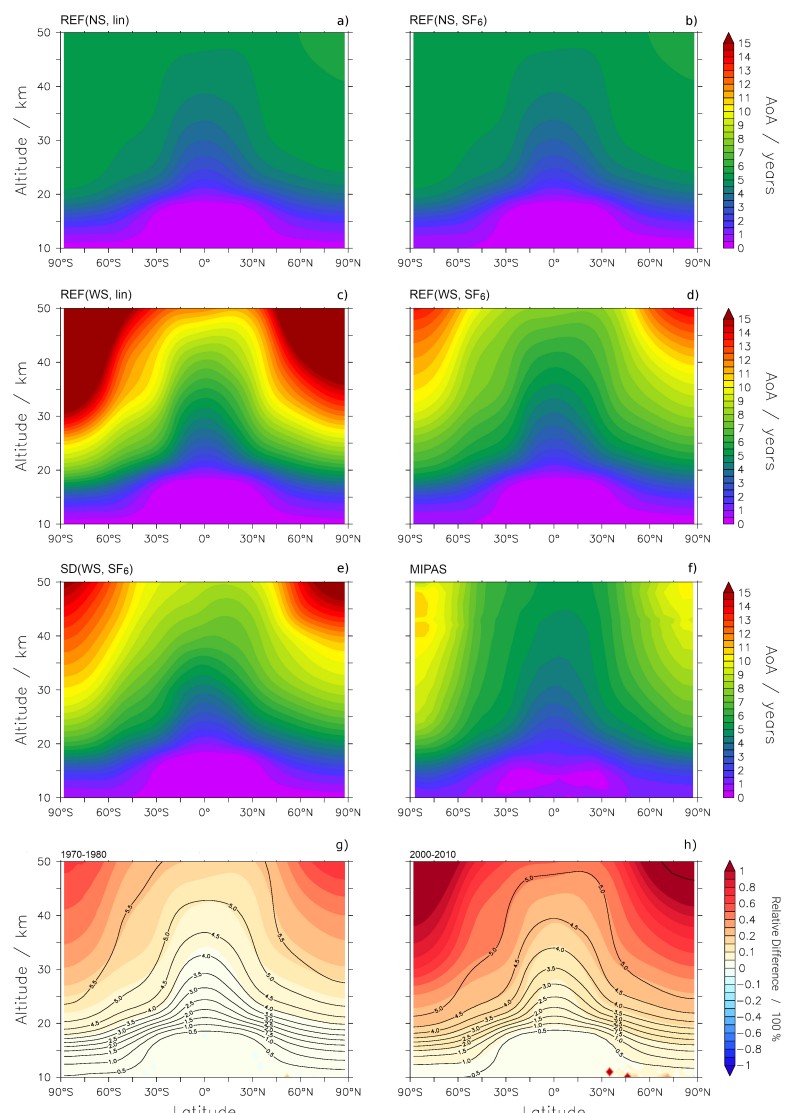

**Figure 3.** AoA climatologies of annual means over 2007-2010. Model AoA from the reference simulation for the different tracers $tr$(NS, lin), $tr$(NS, SF$_6$), $tr$(WS, lin), and $tr$(WS, SF$_6$) is shown in panels a) through d), respectively, while AoA ($tr$(WS, SF$_6$)) from the specified dynamics simulation is shown in panel e). MIPAS (Stiller, 2021) AoA can be seen in panel f). The relative difference between AoA(WS, SF$_6$) and AoA(NS, SF$_6$) from the reference simulation for the period 1970-1980 and 2000-2010 are shown in panels g) and h), respectively, and is calculated using $\frac{\text{AoA(WS,SF}_6)-\text{AoA(NS,SF}_6)}{\text{AoA(NS,}SF_6)}$. The black contours depict AoA(NS, SF$_6$)$_{REF}$ for the respective time period.

to considerably younger AoA in the middle to upper stratosphere. There are, however, good reasons to believe that the most recent MIPAS data are improved compared to the previous ones: the spectroscopic data used are far better characterised than the previous ones (Harrison, 2020), and the new AoA data from MIPAS agree significantly better with independent measurements

than the previous version, in particular at higher altitudes (Stiller et al., 2020; Stiller, 2021). On the other hand, free-running EMAC simulations generally have a too weak Antarctic polar vortex (see Jöckel et al., 2016), which is, however, stronger than that of the reference simulation. Therefore, the more stable vortex in the SD simulation leads to enhanced isolation and ageing of polar stratospheric air, especially in the Southern Hemisphere during austral spring (see Supplement, Fig. S4, for details). This could somewhat resolve the issue for the comparison with the previous MIPAS AoA version (see Stiller et al., 2012; Haenel et al., 2015, for further details); for the present MIPAS version, however, the discrepancies in the high altitudes and latitudes with the EMAC SD simulation are even larger than with the REF simulation. The comparison of Fig. 3e) with Fig. 3f) illustrates that the model cannot reproduce the tropical pipe and exhibits too much horizontal mixing, or too slow upwelling. This could explain why the model does not reproduce the constant AoA with height in the mid-latitudes. A detailed assessment of both the satellite data and the model simulations is necessary to resolve these discrepancies. In the model, dynamical effects like the strength of the polar vortex or the gravity wave parameterisation can play important roles for the downwelling strength. Moreover, various processes of the chemical $SF_6$ removal can be revised and/or parameterised differently. Here, we showed that $SF_6$ sinks have the potential to resolve the differences between simulated and observed climatologies of AoA and that EMAC AoA lies within the uncertainties of MIPAS AoA throughout the atmosphere. We therefore consider our simulations suitable for studying the temporal evolution of AoA.

### 3.4 Apparent age of air trends

In this section, we analyse the EMAC AoA trends and compare them with observation-based AoA trends. Fig. 4 shows the AoA time series and linear regressions from the REF and the SD simulations as well as from MIPAS observations (Stiller, 2021) and from the $SF_6$ measurements by Engel et al. (2009). As the latter were collected from balloon flights in the Northern Hemisphere mid-latitudes at around 30 km altitude, the EMAC and MIPAS data are also taken from that height and averaged over 30°N to 50°N for consistency.

For better quantification of the trends, Tab. 3 provides the AoA trend values of the EMAC simulations for two periods. The trend of the entire simulation period 1965-2011 is taken for long-term trend assessment and comparison with the measurements by Engel et al. (2009). For the comparison with MIPAS data, the EMAC AoA trends are shown for the period 2002-2011 between 30°N and 50°N, for the realistic tracer $tr(\text{WS}, SF_6)$. The trend calculation follows that of Haenel et al. (2015).

The tracers without $SF_6$ sinks lead to negative AoA trends, which are consistent with the simulated acceleration of the BDC in the course of climate change (e.g. Garcia and Randel, 2008). Positive AoA trends are obtained for all tracers that take $SF_6$ chemistry into account. The trend of $0.19 \pm 0.01$ years per decade in the REF simulation (WS, $SF_6$) is within the limits of the uncertainties of the trend obtained by Engel et al. (2009), who calculated an AoA trend of $0.24 \pm 0.22$ years per decade. This means that in our simulations, the sinks help to reconcile the modelled and the measured AoA trends over the recent decades. Note that Engel et al. (2009) and Engel et al. (2017) also obtained a positive trend for AoA derived from $CO_2$ measurements.

Haenel et al. (2015) calculated MIPAS AoA trends for the period 2002-2012 of $0.25 \pm 0.11$ years per decade for 30°N-40°N at 30 km and $0.24 \pm 0.11$ years per decade for 40°N-50°N at 30 km. For the new MIPAS dataset, the AoA trend is $0.34 \pm$

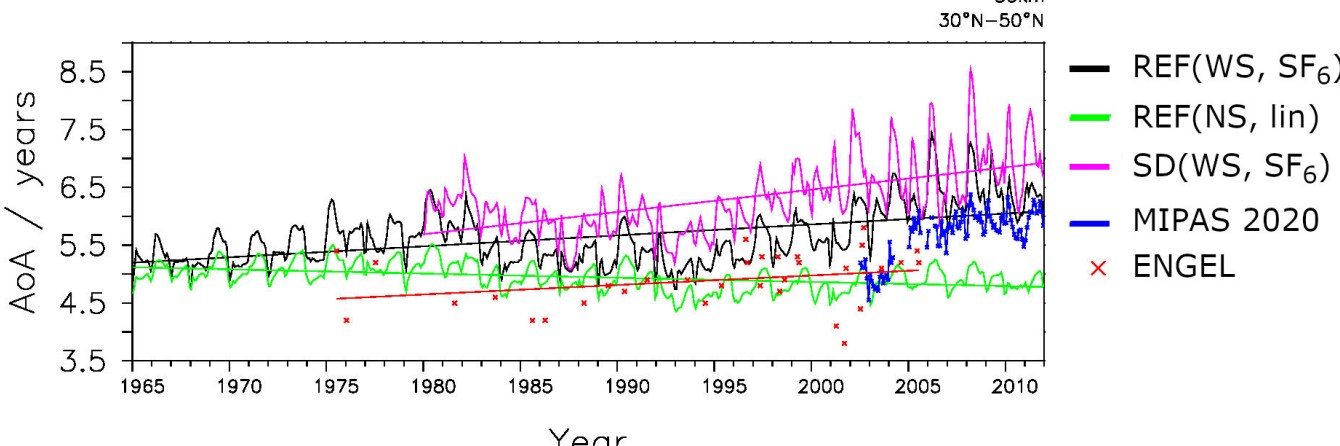

**Figure 4.** AoA time series and linear regressions calculated at 30 km averaged over 30°N-50°N. EMAC AoA from $SF_6$ with non-linear emissions from the reference simulations is shown in black (REF(WS, $SF_6$)); AoA from the tracer with linear emissions without sinks in green (REF(NS, lin)). AoA from $SF_6$ with non-linear emissions with sinks in the specified dynamics simulation SD(WS, $SF_6$) is shown in pink. AoA from balloon-borne measurements (Engel et al., 2009) and from MIPAS observations (Stiller, 2021) are shown in red and dark-blue, respectively.

**Table 3.** EMAC AoA trends at 30 km averaged over 30°N-50°N. The calculation of the 2002-2011 trends is provided for the two relevant simulations, REF(WS, $SF_6$) and SD(WS, $SF_6$), and follows the methods used in Haenel et al. (2015). The 2002-2011 trends are also provided for the remaining simulations.

| Simulation | 1965-2011 Trend (years/decade) | 2002-2011 Trend (years/decade)[a] |
|---|---|---|
| REF(WS, $SF_6$) | $0.19 \pm 0.01$ | $0.22 \pm 0.12$ |
| REF(NS, $SF_6$) | $-0.06 \pm 0.01$ | $-0.07 \pm 0.06$ |
| REF(WS, lin) | $0.70 \pm 0.03$ | $0.23 \pm 0.22$ |
| REF(NS, lin) | $-0.07 \pm 0.01$ | $-0.06 \pm 0.05$ |
| SD(WS, $SF_6$)[b] | $0.39 \pm 0.03$ | $0.50 \pm 0.13$ |
| CSS(WS, $SF_6$) | $0.11 \pm 0.01$ | $0.19 \pm 0.08$ |
| TS2000(WS, $SF_6$)[b] | $0.23 \pm 0.02$ | $0.07 \pm 0.12$ |
| TS2000(NS, lin)[b] | $-0.00 \pm 0.01$ | $-0.21 \pm 0.05$ |
| MIPAS[c] | | $0.34 \pm 0.13$ |
| Engel et al. (2009)[d] | $0.24 \pm 0.22$ | |

[a] Trend calculated following methods of Haenel et al. (2015)

[b] Trend calculated over 1980 - 2011 at 30 km altitude.

[c] MIPAS trend calculated over the 2002-2012

[d] Trend calculated over 1975 - 2005 between 32°N-51°N and 24 km-35 km

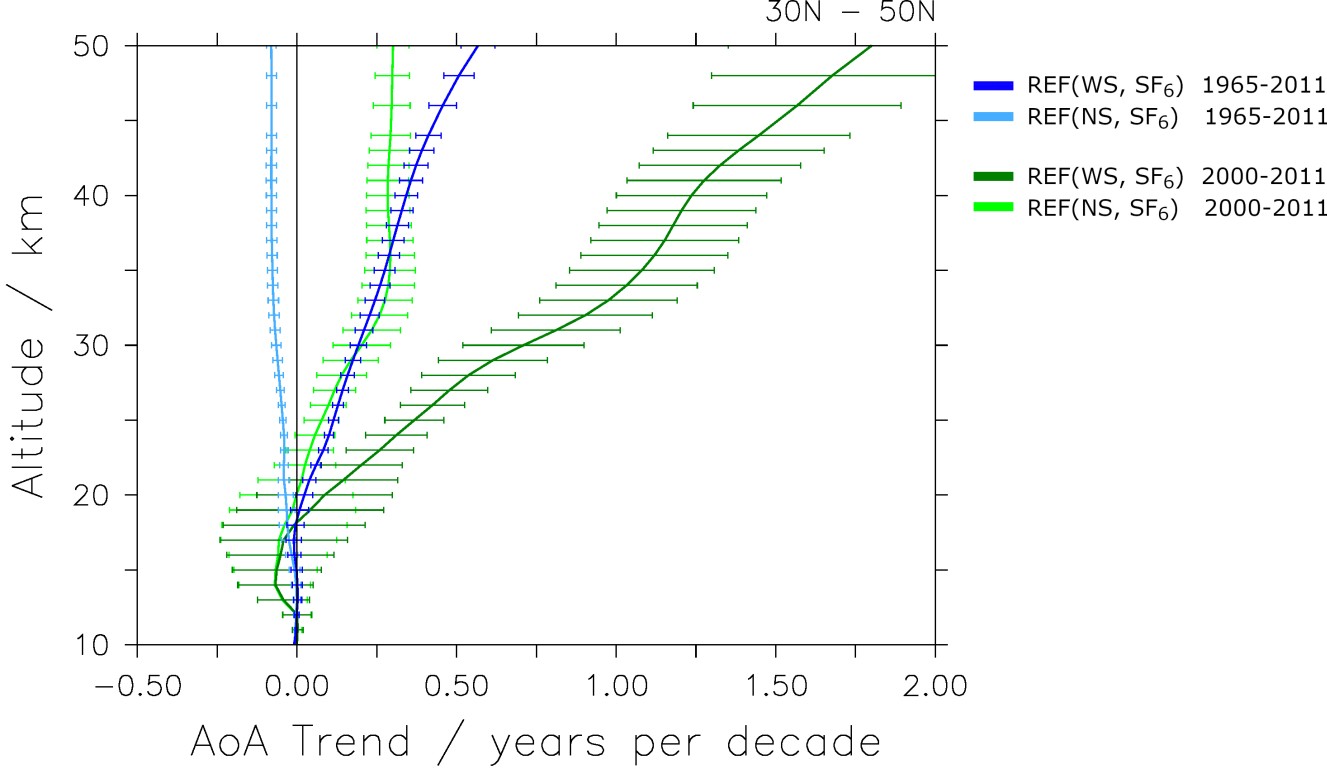

**Figure 5.** Vertical profile of the linear trends of AoA(WS, $SF_6)_{REF}$ and AoA(NS, $SF_6)_{REF}$ over 30°N-50°N, calculated for the two time periods 1965-2011 and 2000-2011. Error bars depict the $2\sigma$ standard deviation of the trend over the respective time period, and the black vertical line denotes the zero-line.

0.13 years per decade. Note that these trends are calculated by applying a bias correction for the discontinuity between the two different observational periods of MIPAS (see Fig. 4), a description of the method can be found in von Clarmann et al. (2010).

Following the trend calculation used in Haenel et al. (2015), the REF(WS, $SF_6$) and SD(WS, $SF_6$) time series show that EMAC AoA bears a good resemblence to that of the new MIPAS retrieval, with an AoA trend of $0.22 \pm 0.12$ and $0.50 \pm 0.13$
330   years per decade, respectively. Consistent with the trend calculation of the MIPAS data, the variability due to the QBO is considered by a respective term in the multivariate linear analysis. However, this measure induces only small differences in the EMAC trend calculations (see Table S1 in Supplementary Information for further details). Note also that the rather short period of the MIPAS observations implies rather large uncertainties in the trends because inter-annual variability can have a large effect on the trend calculations. This is also apparent from the highly variable trend signals in the different simulations,
for example the TS2000(NS, lin) exhibits a strongly negative trend over this period despite no forced long-term trends. This is confirmed by a trend of $-0.00 \pm 0.01$ years per decade over the 1965-2011 period.

As shown in Fig. 5, the strong deviation of $SF_6$-derived AoA trends holds almost throughout the stratosphere. Only below about 20 km altitude for the period from 1965-2011 are the effects of the $SF_6$ sinks smaller than the trend uncertainty. For the

trend in the shorter time period of 2000-2011, effects from the $SF_6$ sinks become significant at about 22 km altitude and higher, which is mainly due to the larger uncertainty stemming from a smaller time period. This means that in the mid-latitudes, the $SF_6$-based AoA trends with and without sinks are not distinguishable from each other up to 20 km and 22 km altitude, depending on the period. Furthermore, the uncertainty in the trend calculated from $SF_6$-based AoA with sinks increases with increasing altitude – this is to be expected, as the effect of the $SF_6$ sinks increases with increasing altitude. The trend of $AoA(NS,SF_6)_{REF}$ over the period 2000-2011 is positive at 25 km and negative over the longer period 1965-2011. However, it is important to note that the time period of about 1 decade implies that the trend is strongly influenced by inter-annual variability (see e.g. Dietmüller et al., 2021). The strong influence of inter-annual variability also explains the difference in trend values in Fig. 5, for which trends are calculated with a simple linear fit, versus the value for a similar period in Table 3. The latter trends were calculated using a regression model taking other variability modes into account to enable a comparison with MIPAS.

### 3.5 Explanations for apparent age of air trends

In this section, we will analyse the EMAC apparent AoA trends, in particular the sign change of the trend when $SF_6$ sinks are switched on. For this, Fig. 6 shows the AoA $(WS,SF_6)$ time series of the sensitivity simulations CSS and TS2000 averaged between 30°N-50°N. For comparison, AoA from the reference simulation (REF(WS, $SF_6$) and REF(NS, lin)) are also included.

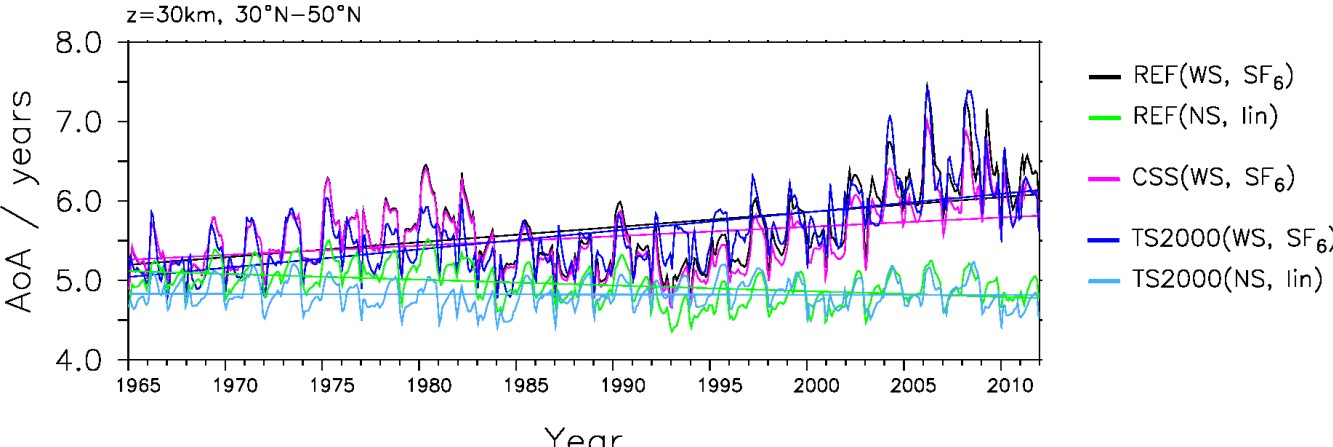

**Figure 6.** AoA (at 10 hPa, averaged over 30°N-50°N) time series and linear regression of sensitivity experiments TS2000(WS, $SF_6$), TS2000(NS, lin), and CSS(WS, $SF_6$) shown in dark blue, light blue, and pink, respectively. Reference simulations REF(WS, $SF_6$) and REF(NS, lin) shown in black and green, respectively.

In the CSS simulation the mixing ratios of the reactant species involved in the sink reactions of $SF_6$ are held constant. The CSS simulation with the realistic tracer (CSS(WS, $SF_6$)) shows an AoA trend of $0.11 \pm 0.01$ years per decade in AoA over the 1965-2011 period. This is lower than the AoA trend in the reference simulation REF(WS, $SF_6$) and means that while the changes in the $SF_6$-depletive substances influence the magnitude of the positive trend, they cannot explain the positive sign of the AoA trend.

The TS2000 simulation is a time slice simulation with climate conditions of the year 2000. AoA of that simulation derived from the realistic tracer (TS2000(WS, $SF_6$)) shows a positive trend of $0.23 \pm 0.02$ years per decade. This is even stronger than the trend of the REF simulation ($0.19 \pm 0.01$ years per decade). By definition, the TS2000 simulation does not feature any changes in its climatic state or in the composition of the atmosphere. This is reflected by the fact that the idealised tracer $tr$(NS, lin) does not show a trend in this simulation (see Tab. 3, TS2000(NS, lin): $-0.00 \pm 0.01$). The temporal increase in apparent AoA rise in the TS2000(WS, $SF_6$) simulation despite no climate changes therefore points to the fact that the $SF_6$ sinks themselves lead to the positive trend. The difference between the TS2000 and the REF AoA trends for both $tr$(WS, $SF_6$) and $tr$(NS, lin) reflect the negative contribution of the accelerating BDC to the trend, which can be seen in the idealised AoA trend from the REF simulation (NS, lin). Overall, neither changes in the $SF_6$ sinks nor changes in the stratospheric circulation due to climate change are responsible for the positive trend found in AoA with sinks. Instead, the results indicate that the sinks themselves can generate a positive trend.

For a complete discussion of the features that can be seen in Fig.6, we now also describe the sudden decrease in AoA(WS, $SF_6$) shortly after 1982 seen in the REF simulation. It is also visible in the two sensitivity simulations (CSS and TS2000). This means that changes in the $SF_6$-depleting substances as well as climate change and volcanic activity (in particular the volcanic erruption of El Chichón in 1982) can be ruled out as a possible cause for the drop. The solar cycle is not taken into account in the timeslice simulation. The timeseries of the realistic $SF_6$ tracer emissions show a stronger increase in the 1980s (not shown). Fritsch et al. (2020) showed that due to this increase the calculation of AoA based on $SF_6$-like tracers is more sensitive to the chosen parameters in the AoA derivation for this time. The drop in AoA we see is caused by this limitation of the derivation from $SF_6$-like tracers.

The increasing variability and trend of AoA in the last two decades, seen in the simulations with mesospheric $SF_6$ chemistry, can be attributed at first order to the $SF_6$ depletion reactions (see Fig.6). In particular, the effect of mesospheric $SF_6$ sinks is stronger with higher $SF_6$ mixing ratios. Subsequent downward transport of $SF_6$ depleted air into the vortex and in-mixing thereof into lower latitudes after the vortex breakup results in apparent older AoA. This can explain that the annual variability increases over time due to the increase in $SF_6$ mixing ratios.

## 4  Theoretical considerations and concept for sink correction methods

The following section focuses on the theoretical examination of the link between $SF_6$ sinks and positive AoA trends. First, we will show from a theoretical standpoint that $SF_6$ sinks with constant destruction rates lead to a positive trend in AoA. Based on the theoretical considerations and on the model data we will then discuss the possibilities of a correction of AoA derived from $SF_6$ data for the effects of the sinks. Many observational mean AoA estimates are based on measurements of $SF_6$, and given that the relevance of the sinks of the mean AoA estimates increases over time, a correction method is required to obtain unbiased information on stratospheric transport strengths.

The aforementioned link between the positive AoA trend and mesospheric $SF_6$ depletive chemistry, based on Hall and Waugh (1998), is illustrated below and follows the mathematical formulations put forward by Hall and Plumb (1994) and

Schoeberl et al. (2000). To allow for analytical expressions, we will only consider the case of a linearly increasing tracer here, and further assume that the lifetime of $SF_6$ is constant in time. We consider a tracer $\chi(t)$ experiencing relative loss $e^{-t/\tau}$ with time-constant lifetime $\tau$. Note that $\tau$ is equivalent to the inverse of the loss rate. We denote the reference mixing ratio as $\chi_o(t)$ and assume a constant growth rate of $\Delta\chi_{oo}$ and write

$$\chi_o(t) = \Delta\chi_{oo} \cdot t \tag{3}$$

For any location we can then express the tracer mixing ratio as

$$\chi(t) = \int_{t'=0}^{\infty} \chi_o(t-t') \cdot e^{(-t'/\tau)} \cdot G(t')dt' \tag{4}$$

where $t'$ denotes the transit time. $G(t')$ represents the Green's function and is equivalent to the AoA spectrum. Inserting (3) into (4) gives us

$$\chi(t) = \Delta\chi_{oo} \cdot t \int_{t'=0}^{\infty} e^{(-t'/\tau)} \cdot G(t')dt' - \Delta\chi_{oo} \int_{t'=0}^{\infty} t' e^{(-t'/\tau)} \cdot G(t')dt' \tag{5}$$

The expression under the first integral corresponds to the arrival time distribution $G^*(t') = exp(-t'/\tau)G(t')$. This represents the transit time distribution of a chemically depleted tracer with lifetime $\tau$ (see e.g. Plumb et al. (1999), however note that $G^*$ is not normalised in our definition, following Engel et al. (2017)). Note that $\tau$ is a measure of the lifetime an air parcel experiences on its path of length $t'$ and is referred to the path-integrated lifetime (not to be confused with the local lifetime). The second integral in (5) is the first moment of the arrival time distribution $G^*(t')$, i.e. the mean arrival time and is denoted as $\Gamma^*$. Using these terms, (5) can be expressed as

$$\chi(t) = \Delta\chi_{oo} \left( t \cdot \int_{t'=0}^{\infty} G^*(t')dt' - \Gamma^* \right) \tag{6}$$

For the tracer without sinks (here referred to as the passive tracer $\chi_p(t)$) the integral over $G^*(t')$ equals 1, and $\Gamma$ represents the mean AoA. Eq. 6 can then be rearranged to

$$\Gamma = t - \frac{\chi_p(t)}{\Delta\chi_{oo}} \tag{7}$$

giving the common expression to derive mean AoA from a linear increasing tracer.

In the case of a tracer with sinks, $\chi_s(t)$, the calculation of (apparent) mean AoA based on Eqs. 6 and 7 becomes:

$$\tilde{\Gamma} = t - \frac{\chi_s(t)}{\Delta\chi_{oo}} = t - \left( t \cdot \int_{t'=0}^{\infty} G^*(t')dt' + \Gamma^* \right) = t \left( 1 - \int_{t'=0}^{\infty} G^*(t')dt' \right) + \Gamma^* \tag{8}$$

The change of apparent mean AoA with time can then be expressed as:

$$\frac{\partial\tilde{\Gamma}}{\partial t} = 1 - \int_{t'=0}^{\infty} G^*(t')dt' \tag{9}$$

In the case of a passive tracer (i.e. a tracer without sinks), $G^*(t')$ is equal to the age spectrum $G(t')$ and thus its integral equals 1. Consequently the AoA trend is zero in the absence of a trend in transport (constant $G$). However, for the depleted tracer (with sinks) a positive linear trend in the apparent mean AoA ($\tilde{\Gamma}$) is induced by the sinks, even if the circulation strength (i.e. $G(t')$) and the lifetimes $\tau(t')$ are constant. This is consistent with the results shown in the previous sections. In particular, we showed a positive trend linearly increasing $SF_6$-like tracer in the time slice experiment (TS2000, see e.g. Table 2), which satisfies all conditions made above. In the reference simulation the BDC is accelerating over time, so that mean AoA from the passive tracer is decreasing. However, in our simulations the positive trend induced by the sinks overcompensates the impact of the BDC acceleration. It is important to note that the above is valid for a linearly increasing tracer, and that variable growth rates will modify the influence of $SF_6$ on apparent AoA.

A correction of the mean AoA estimate for the effects of the sinks would require knowledge of the (integrated) arrival time distribution $G^*$ (or, equivalently, the transit-time dependent path integrated lifetimes $\tau(t')$), a quantity that is not readily available and possibly non-linearly dependent on $\Gamma$. As a thought experiment, and with the aim to derive an analytical concept for the correction of mean AoA for the sinks, we make the hypothetical assumption that the age spectrum is represented by a single, average path, i.e. we assume the age spectrum is a delta-function $G(t') = \delta(\Gamma)$. With this assumption, the tracer mixing ratio of the depleted tracer (Eq. 5) simplifies to:

$$\chi_s(t) = \Delta\chi_{oo} \cdot e^{-\frac{\Gamma}{\tau_{eff}}} \cdot (t - \Gamma) \tag{10}$$

where $\tau_{eff}$ is the path-integrated lifetime along this single path. To avoid confusion with either the local lifetime or the averaged transit-time dependent path-integrated lifetime, we will refer to $\tau_{eff}$ as the 'effective' lifetime.

The apparent mean AoA calculated from the depleted tracer $\chi_s(t)$ follows from Eqs. 7 and 10:

$$\tilde{\Gamma} = t - \frac{\chi_s(t)}{\Delta\chi_{oo}} = t\left(1 - e^{-\frac{\Gamma}{\tau_{eff}}}\right) + e^{-\frac{\Gamma}{\tau_{eff}}} \cdot \Gamma \tag{11}$$

Given that the mean age $\Gamma$ is about 2 orders of magnitude lower than the lifetime $\tau_{eff}$, we can approximate $e^{-\frac{\Gamma}{\tau_{eff}}} \approx 1 - \frac{\Gamma}{\tau_{eff}}$, and thus $\tilde{\Gamma} = \Gamma\left(1 + \frac{t}{\tau_{eff}}\right) - \frac{\Gamma^2}{\tau_{eff}}$. Again for small $\Gamma$ compared to $\tau_{eff}$, the second term can be neglected so that

$$\tilde{\Gamma} \approx \Gamma\left(1 + \frac{t}{\tau_{eff}}\right) \tag{12}$$

In general, $\tau_{eff}$ will be dependent on $\Gamma$. However, if the effective lifetime were constant, the apparent mean age $\tilde{\Gamma}$ would be linearly related to the actual mean AoA. The slope between the latter two then is merely a function of the effective lifetime and increases linearly over time $t$. Thus, for constant circulation ($\Gamma$) and effective lifetime $\tau_{eff}$, a linear trend of strength

$$\frac{\partial\tilde{\Gamma}}{\partial t} = 1 + e^{-\frac{\Gamma}{\tau_{eff}}} \approx \frac{\Gamma}{\tau_{eff}} \tag{13}$$

is induced.

Under these assumptions a relatively simple, linear relation between the true mean AoA ($\Gamma$) and the apparent AoA ($\tilde{\Gamma}$) is obtained. While the assumptions will clearly be violated in the model, we investigate, based on data from our model simulations, whether the violations might effectively be small enough so that the linear relation still holds. In particular, the linearly

increasing tracers from the time slice simulation (TS2000) with constant circulation strength satisfy the initial assumptions (linear increase and constant circulation), and we can test whether the approximations of a single average path hold.

Fig. 7 shows a quasi-linear relation between the ideal AoA and AoA derived from the linearly increasing $SF_6$-like tracer for

mean AoA values below approximately 4 years. The slope increases over time (as apparent from the transition from yellow to red colors in Fig. 7), consistent with the simplified theoretical considerations for a constant effective lifetime $\tau_{eff}$ (see Eq. 12). However, for mean AoA values above about 5 years, this quasi-linear regime does not hold anymore. We instead find exponential growth of $SF_6$-based apparent mean AoA. Those older mean AoA values are closer to the region of depletion in the mesosphere and a constant effective lifetime (as assumed for the linear regime) does not hold anymore. Rather, we can

expect $\tau_{eff}$ to be strongly dependent on $\Gamma$.

Based on Eq. 11 (or its linear approximation Eq. 12), we can estimate the effective lifetime from the model data of $\Gamma$ (from the passive tracer) and $\tilde{\Gamma}$ (from the linearly increasing $SF_6$-like tracer). The resulting lifetime is shown in Fig. 8 as a function of the ideal mean AoA. Consistent with the previous results, $\tau_{eff}$ varies comparatively little with $\Gamma$ for a range between 1 and 4 years. Fig. 7 also includes the theoretical values of $\tilde{\Gamma}$ according to Eq. 11 for $t = 40$ years and when using either a constant

mean value for $\tau_{eff}$ of 140 years (obtained for $\Gamma$=3.3 years), or the $\Gamma$-dependent values. The approximation of a constant effective lifetime matches the model data for ages below about 4 years reasonably well, but for older air, the assumption of a constant $\tau_{eff}$ obviously fails. When using the $\Gamma$-dependent value for $\tau_{eff}$, on the other hand, the exponential increase of $\tilde{\Gamma}$ is overall well captured. This justifies the applicability of the simplified assumptions of one average path (Equations 11 to 12).

Overall, the results derived in this section indicate that a correction of observational $SF_6$-derived mean AoA for the effects

of chemical sinks is likely possible by applying a time-dependent linear correction function. This linear relation between AoA from the ideal and from the chemically depleted $SF_6$ tracer holds for mean ages below about 4 years. Here, we show this relation for the linearly increasing tracer in northern mid-latitudes. However, further analysis indicates that this linear relation also holds for the realistic $SF_6$ tracer with non-linear growth over time (not shown). Furthermore, the linear relation seems to be nearly identical for different latitude bands (not shown), which is a very promising property for future applications of a

correction method.

We emphasize that the strength of the $SF_6$ sink in our model simulations is not well enough constrained to properly establish such a correction function. Deriving suitable values for the linear relation between $\Gamma$ and $\tilde{\Gamma}$ (and thus the effective lifetime $\tau_{eff}$) should be obtained by means of observational data. This could be achieved by using simultaneous measurements of $SF_6$ and other age tracers, as previously shown by Leedham Elvidge et al. (2018) and Adcock et al. (2021). Furthermore, the concept

needs to be evaluated vigorously on model data to assess its errors and limitations.

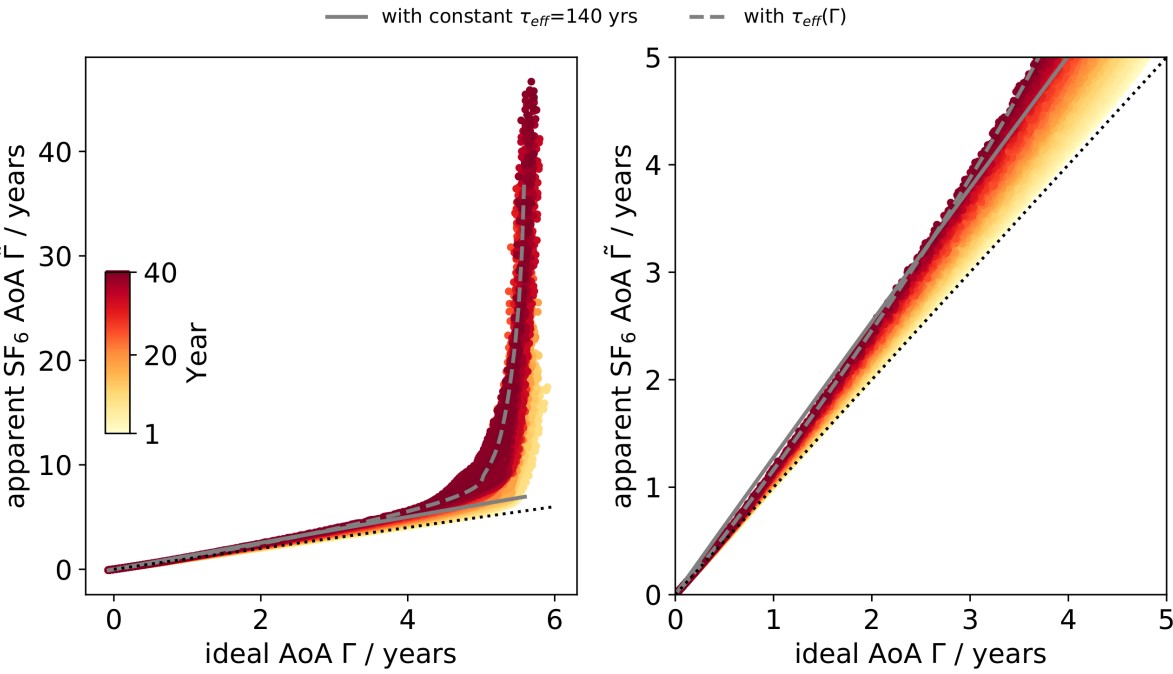

**Figure 7.** Mid-latitude (30°N-50°N) averaged mean AoA from the ideal tracer plotted against the apparent mean AoA derived from the linearly increasing SF$_6$-like tracer from the TS2000 simulations, for years ranging from year 1 (yellow) to year 40 (dark red). The right panel shows the same as the left, but limited to ages below 5 years. The gray lines are estimations based on Eq.11 for t=40 years, with either a constant effective lifetime of 140 years (dashed) or with the effective lifetime dependent on $\Gamma$ (as shown in Fig.8).

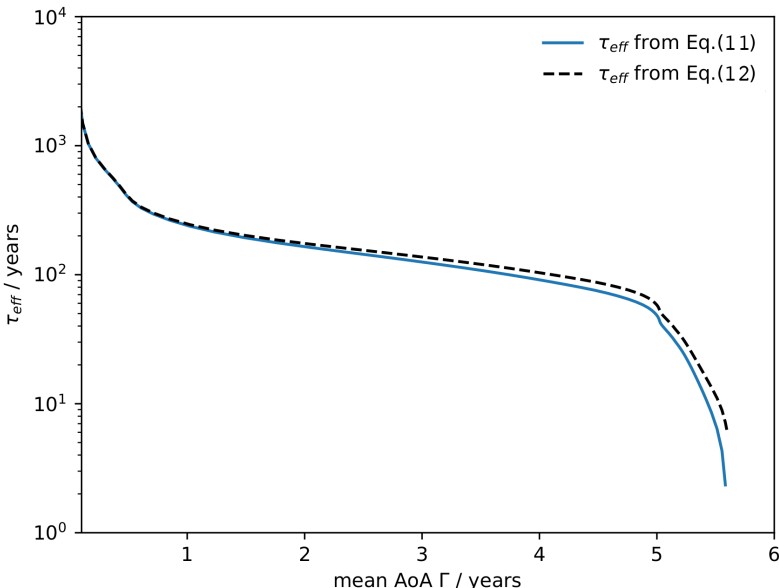

**Figure 8.** Effective lifetime derived from average mid-latitude (30°N-50°N) ideal mean AoA and apparent mean AoA (from the linearly increasing $SF_6$-like tracer) from the TS2000 simulations. The calculation of $\tau_{eff}$ is based on Eq.11 (blue solid) and its linear approximation Eq.12 (black dashed).

## 5 Summary and conclusions

Disagreements with regards to stratospheric AoA and particularly its trends between model simulations and observations have been raising many questions by scientists for more than a decade now. AoA from observations is mostly older than AoA from model simulations and models simulate a decrease in AoA over recent decades, whereas trend estimates from observational data
report a non-significant positive trend. In agreement with our results, previous studies (see, e.g. Kouznetsov et al., 2020) showed that the chemical sinks strongly influence $SF_6$-derived AoA in terms of absolute values and decadal changes. We investigate for the first time how longer-term trends are affected in a consistent manner and investigate the different contributions from circulation changes, changes in abundance of reaction partners, and trends induced by constant destruction rates. To make this step towards understanding the reasons for the discrepancies, we thus study the impact of mesospheric $SF_6$ sinks on AoA
climatologies and trends using the chemistry climate model EMAC (Jöckel et al., 2010; Jöckel et al., 2016) with the SF6 submodel (Reddmann et al., 2001). This submodel allows for explicit calculation of $SF_6$ sinks and we applied a correction for the non-linear growth of $SF_6$ in the calculation of AoA (Fritsch et al., 2020).

The EMAC $SF_6$ mixing ratio profiles show good agreement with balloon-borne measurements as well as with satellite observations. Some of the differences between the model and observations are within the uncertainty range of the observations.
However, reasons for the quantitative differences in the high latitudes and altitudes can also be found in deficiencies in the representation of the $SF_6$ sinks in the model or in the dynamics simulated by the model. The EMAC reference simulation yields a global stratospheric $SF_6$ lifetime of 2100 years, varying between 2500 years and 1900 years for the simulation period. This value lies within the range of 600-2900 years provided by the model study of Kouznetsov et al. (2020) and below the value 3200 years calculated by Ravishankara et al. (1993). Kovács et al. (2017) and Ray et al. (2017) recently found somewhat lower $SF_6$
lifetimes of 1278 years and 850 years, respectively. Although this shows that large uncertainties still exist in determining the $SF_6$ lifetime, these results also confirm that the EMAC $SF_6$ depletion mechanisms are reasonable. In our transient simulations (REF and PRO), the $SF_6$ lifetimes vary by about 25 % following the abundances of the reactant species, basically resembling the pattern of the stratospheric ozone concentration. This behaviour, however, may be a result of the fact that several effects that potentially influence $SF_6$ lifetime variability are not implemented in full detail.

The inclusion of $SF_6$ sinks translates into apparent older stratospheric air. When $SF_6$ sinks are enabled, EMAC AoA therefore compares better with MIPAS satellite observations. In the tropics, overall good agreement can be found, but the results also indicate that EMAC does simulate a too broad, less isolated tropical pipe. In polar regions, however, EMAC AoA is higher than MIPAS AoA. In comparison with the previously published MIPAS data (Stiller et al., 2012; Haenel et al., 2015), the EMAC AoA in the polar regions was actually too low. More research on both models and observations is necessary to resolve these
remaining discrepancies.

In this study we show the effect of $SF_6$ sinks on the tracer-derived AoA trend for both longer time series as well as for the short time series corresponding to the MIPAS time frame, whereby issues in variability are associated with the latter. Without $SF_6$ sinks, EMAC shows a negative AoA trend over 1965-2011. This is consistent with the simulated acceleration of the Brewer-Dobson circulation resulting from climate change (see e.g. Garcia and Randel, 2008; Butchart et al., 2011; Eichinger

et al., 2019). The inclusion of chemical $SF_6$ sinks leads to positive AoA trends throughout the stratosphere (except the tropical lower stratosphere below 50 hPa) in our simulations, which in turn is consistent with the positive AoA trend derived from MIPAS in the Northern Hemisphere (Haenel et al., 2015). Moreover, the $SF_6$ sinks help to improve the agreement of our model results with the AoA derived from the balloon-borne in situ measurements by Engel et al. (2009), from which a (non-significant) positive AoA trend was obtained. However, this only accounts for $SF_6$-derived AoA, and our results cannot help

explain the positive trend of the $CO_2$-derived AoA in Engel et al. (2009, 2017). Furthermore, it has recently been shown that the AoA trend derived in Engel et al. (2009, 2017) was likely overestimated due to non-ideal parameter choices in the calculation of AoA (Fritsch et al., 2020).

Our sensitivity studies quantified that the positive AoA trends are neither a result of climate change, nor of changes in the substances involved in $SF_6$ depletion. The $SF_6$ sinks themselves are the reason for the increase in apparent AoA. The reason

for that is the temporally increasing influence of the chemical $SF_6$ sinks on AoA. In our simulations, this effect overcompensates the effect of the simulated acceleration of the stratospheric circulation leading to a net increase of AoA. Due to various sources of uncertainties, this result bears quantitative leeway and has to be assessed in finer detail. But nevertheless, for now we can conclude that $SF_6$ sinks have the potential to explain the long-lasting AoA trend discrepancies between models and observations. Furthermore, we put forward a first approach towards a method for $SF_6$ loss correction, which in the future shall

be further developed and applied on observational data. From our first analyses, we can conclude that a linear correction (that is dependent on both time and effective lifetime of $SF_6$) can likely be applied to AoA values up to 4 years. However, further studies with more comprehensive approaches are required for a precise quantification of these values.

*Code availability.* The Modular Earth Submodel System (MESSy) is continuously developed and applied by a consortium of institutions. Use of MESSy and access to the source code is licensed to all affiliates of institutions that are members of the MESSy Consortium. Institutions

can become a member of the MESSy Consortium by signing the MESSy Memorandum of Understanding. More information can be found on the MESSy Consortium website (http://www.messy-interface.org, last access: 27 March 2020). The exact code version used to produce the simulation results is archived at the German Climate Computing Center (DKRZ) and can be made available to members of the MESSy community upon request.

*Data availability.* The simulation results are archived at DKRZ and are available upon request. MIPAS $SF_6$ and AoA data are available from

535 G. Stiller on request.

*Author contributions.* SL, RE and HG designed the study. SL analysed the data with support from RE, HG and FF. RE performed the simulations. TR and SV implemented the SF6 submodel in MESSy. GS and FH provided the MIPAS data analyses. SL and RE drafted the paper and all authors helped with discussions and with finalising the manuscript.

*Competing interests.* The authors declare no competing interests.

*Acknowledgements.* This study was funded by the Helmholtz Association under grant VH-NG-1014 (Helmholtz-Hochschul-Nachwuchsfor-schergruppe MACClim). RE acknowledges support by GA CR under grand nos. 16-01562J and 18-01625S. This work used resources of the Deutsches Klimarechenzentrum (DKRZ) granted by its Scientific Steering Committee (WLA) under project ID bd1022. Moreover, we thank Eric Ray and Andreas Engel for providing observational data and Axel Lauer for his helpful suggestions as well as Rostislav Kouznetsov and Eric Ray (again) for their thorough reviews of the paper.

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
