# Peer review of "The impact of SF6 sinks on age of air climatologies and trends"

_Atmospheric Chemistry and Physics, 2021_

## Author Comment (AC1)

**Reply to Referee #1**

*This paper explores the effects of SF6 loss on the mean age values and trends derived from SF6 model output and observations. The conclusion, that SF6 loss can reverse the sign of the mean age trend over recent decades compared to mean age from a tracer without loss, is important in helping us understand the long running discrepancy between modeled and measured trends. The explanation is actually incredibly straightforward, that the higher the concentration of a species in the atmosphere, the more absolute loss will occur and therefore the bias will grow over time. It's kind of hard to believe none of us studying this topic hadn't thought of this possibility before, but there you go. Nice work by the authors of this paper to recognize the importance of SF6 loss and quantify the effect.*

*There are still outstanding issues, the trend in CO2 derived mean age is one of them and the SF6 lifetime derived in the EMAC model is another since it seems to be higher than other recent estimates. Nevertheless, this paper is an important step forward in our understanding of mean ages derived from observations and how models can be used to help put them in context. My main comments described below revolve around how best to use this information to help us make the observationally derived SF6 mean ages more accurate. It doesn't make sense to cast the measurements aside when it comes to mean age estimates so we need to be careful to frame the current understanding in a more inclusive way.*

*I do recommend this paper be published with consideration of the comments below.*

**Main comments**

*It would be really nice to see a plot of latitude vs. altitude trend differences between REF(WS, SF6) and REF(NS, SF6). I realize you've focused the trend analysis on the locations of the Engel et al. papers in the 4-6 year mean age range, and that's important to see. But at what mean age does the trend discrepancy emerge from the uncertainty to become significant? How does this vary with location? You hint at it in Figure 1 where the NS and WS profiles diverge but it would be good to see more. I think this is important for the community to know what age ranges we can reliably use SF6 as a mean age tracer. For instance, in my 2014 paper I showed mean age trends at four altitude ranges and in the bottom layer, where mean ages were 2-3 years, the SF6 measurement trends agreed within uncertainties with the modeled mean age trends. Is this expected based on your work?*

**Reply #1:**
Thank you for your comment, and this is a very good suggestion. We included the relative difference of both the mean AoA climatology as well as the trends as latitude vs altitude plots. For the climatologies, we included the relative difference between AoA from REF(WS, SF6) and REF(NS, SF6) for two periods, namely 1970-1980 and 2000-2010. These two figures have been added to Figure 3. The caption has been amended to include:
"The relative difference between AoA(WS, SF6) and AoA(NS, SF6) from the reference simulation for the periods 1970-1980 and 2000-2010 are shown in panels g) and h), respectively. The black contours depict AoA(NS, SF6)REF for the respective time period. "
and the following text has been added to line 229:
"The relative effect of the sinks on AoA derived from the SF6 tracers with non-linear growth can be seen in Figs. 3g) and h). Mean AoA values derived from SF6 in the early period (1970-1980) are moderately affected by the sinks, with a difference of around 20-25% in the polar middle stratosphere and above (i.e., for mean AoA values above 5.5 years). Differences are small (less than 10%) for mean AoA values below 4 years. However, as will be discussed (see Section 3.5), the effect of the SF6 sinks increases over time and for the later period (2000-2010), mean AoA derived from SF6 is affected by the sinks considerably. Differences greater than 20% can be seen in Fig. 3h) for

AoA above about 3 years, and in the extratropical lower stratosphere, differences are larger than 10% even for mean AoA values of 2 years and above."

Furthermore, we plotted the latitude vs altitude trend differences between REF(WS, SF6) and REF(NS, SF6) and added this to the revised manuscript (Figure 5). The caption reads as:

"Relative difference between the trends of AoA(WS, SF6) and AoA(NS, SF6) over the time periods (a) 1965-2011 and (b) 2000-2010. The black contours depict the trend of AoA(NS, SF6) over the respective years. Dotted regions indicate where the relative trend difference is not significant, i.e. the statistical significance lies below the 5% threshold."

This figure indicates that the long-term trend (over 1965-2011) for SF6-based AoA with sinks deviates beyond 100% from the long-term trend of SF6-based AoA without sink consideration, which implies that SF6-based AoA is unsuitable for AoA trend calculations in the whole stratosphere. These differences are generally less pronounced for the shorter period of 2000-2010, and the relative trend differences in AoA(WS, SF6) and AoA(NS, SF6) lie under 100% in the tropical and extratropical lower stratosphere.

This has been added as a paragraph at the end of Section 3.4, in line 294:

"As shown in Fig. 5, the strong deviation of SF6-derived AoA trends holds almost throughout the stratosphere: Figure 5a) indicates that the long-term trend (over 1965-2011) for SF6-based AoA with sinks deviates beyond 100% (i.e., by a reversal of sign) from the long-term trend of SF6-based AoA without sink consideration, which in turn implies that SF6-based AoA is unsuitable for AoA trend calculations in the whole stratosphere. These differences are generally less pronounced for the shorter period of 2000-2010 (see Fig. 5b), and the relative trend differences in AoA(WS, SF6) and AoA(NS, SF6) lie under 100% in the tropical and extratropical lower stratosphere."

*As a corollary to your findings, couldn't a correction to SF6 mixing ratios be made in the calculation of mean age to account for loss? This would be similar to the way CO2 is adjusted for production from CH4 oxidation before calculating mean age. This correction would have to vary with time and location, which you could derive from the differences between your REF(NS, SF6) run and the MIPAS data.*

**Reply #2:**
Thank you for your thoughts on this – while this topic will require more in-depth analysis than there is room for in this paper, and we aim to study this in more detail in a follow-up study, we have now extended the mathematical discussion in Section 3.5 (starting at line 329) to its own Section 4: "Theoretical considerations and concept for sink correction methods". In this section, first analyses of a possible corrective approach to account for SF6 loss are put forward, and we conclude that it might be possible to apply a linear correction to AoA of up to 4 years.

As such, we have amended line 19 at the end of the Abstract to read as:

"We conclude the study with a first approach towards a correction to account for SF6 loss and deduce that a linear correction might be applicable to values of AoA of up to 4 years."

The final paragraph has been extended in line 407 in the Conclusion and now reads as:

"Furthermore, we put forward a first approach towards a method for SF6 loss correction, which in the future shall be further developed and applied on observational data. From our first analyses, we can conclude that a linear correction (that is dependent on both time and effective lifetime of SF6) can likely be applied to AoA values up to 4 years. However, further studies with more comprehensive approaches are required for a precise quantification of these values."

**Specific comments**

*Line 6: I would add 'in the NH extratropical middle stratosphere' after 'positive trend'. Or something similar to give some reference to the Ploeger et al. (2015) and Stiller et al. (2017) hemispheric shift effect on the age trends that you mention in the introduction. Also, in Ray et al. (2014) I showed that the balloon age trends are negative in the lower extratropical stratosphere.*

**Reply #3:**
Thank you, we have incorporated your comments in the Abstract. The sentence we added reads:
"Satellite observations show much older air than climate models and whilemost models compute a clear decrease of AoA over the last decades, a thirty-year timeseries from measurements shows a statistically non-significant positive trend in the NH extratropical middle stratosphere."
We also amended the section in the Introduction to read as:
"… show a (statistically non-significant) positive trend (note that Ray et al. (2014) also shows negative balloon AoA trends in the lower extratropical stratosphere)."

*Figure 1: It seems like you could do a better job of averaging model data to match the March 2000 balloon profile location. This balloon flight was clearly in the vortex as shown in Figs. 2 and 3 in Ray et al., 2017 with equivalent latitudes from 68-75N. By averaging model locations at all longitudes and latitudes as low as 60N you are mixing locations in and out of the vortex. It's relevant to the lifetime estimates to see how the polar vortex model and measurement profiles compare since this is the region most impacted by the SF6 loss.*

**Reply #4:**
Thank you for your constructive comment. We have examined the polar vortex in the SD simulation for 5[th] March 2000 and find that it is slightly displaced and centered over Northern Europe. To ensure that we show the SF6 profile based on air masses from within the vortex, we changed the latitude and longitude selection to 65N-80N and 0E-100E. We have attached a figure of SF6 as a function of latitude and longitude at 30km at the end of this reply (see below).
Figure 1 has been changed accordingly and updated in the revised manuscript.
Lines 171-173 have been updated to read as follows:
"To ensure that the SF6 profile is based on air masses from within the vortex, modelled SF6 values are averaged over 0E-100E and 65N-80N, which corresponds to the area of the vortex core for the given day. The standard deviation of SF6 for 0E-100E averaged over the respective latitude range is shown as error bars."

[Figure]

*Line 130: 'approximately' is misspelled, change 'turned out to be' to 'are'*
*Line 199: 'definition of Braesicke…'*
*Figure S4: No y-axis labeling.*
*Figure S5: In the caption you state 'Values averaged over the region 30-50N and 2007-2010'. I think the 30-50N part is not needed here since these are latitude vs altitude plots.*

**Reply #5:**
Thank you for pointing out these errors, we have incorporated your comments in the relevant sections.

**Reply to Referee #2**

**General comments:**

*1. The introduction suggests that main objective of the study is to reveal the cause of discrepancy between modelled and observed AoA and their trends. If it is the case, it should be stated explicitly. The lack of clearly formulated objectives and research questions makes it difficult to understand e.g. the model experiment design and justification for the choice of specific setups for the model experiments.*

**Reply #1:**
Thank you for your comment. The main objective of our study is to further the general understanding of the effect of SF6 sinks on the derivation of SF6 based AoA, namely the resulting discrepancies between modelled and observation-based AoA, and investigate the effect on the long-term trend. To make this clearer, we added the following to line 64:
"… with the aim to understand the effects of SF6 sinks on tracer derived AoA and its long-term trends. Specifically, we calculate for the first time the effect of the sinks on the long-term trend of AoA."

*2. Further, the introduction states that "a comprehensive explanation for the trend differences between models and observations is still missing". The effect of the SF6 destruction on the apparent AoA has been already pointed by Waugh and Hall (2002, Sec 3.2) and addressed by Kouznetsov et al. (2020), who has simulated the effects of the mesospheric sink of SF6, and concluded that "The apparent over-ageing introduced by the sink is large and variable in space and time. Moreover, the over-ageing due to the sink increases as the atmospheric burden of SF6 grows". (For more details, please refer to Sec.6.3 of the latter paper.) Therefore, it should be clearly stated what makes the existing explanations non-comprehensive.*

**Reply #2:**
Thank you for your comment. Kouznetsov et al. (2020) carried out simulations to study the SF6 sinks in a chemistry transport model, while our study uses a chemistry climate model to examine the effects on the long-term trend of AoA. We also provide a correction of the non-linear growth of SF6 in the calculation of AoA in our chemistry climate model.
As such, we have amended line 49, it now reads:
"… In a study based on a chemistry transport model, Kouznetsov et al. (2020) showed that changes in SF6-derived apparent AoA over one decade are highly influenced by the SF6 sink, and can even turn positive. However, a comprehensive understanding of what contribution the individual effects have on the AoA trend depending on altitude and latitude is still missing."
Furthermore, line 51 has been amended to read:
"… SF6 sinks lead to older apparent AoA (see e.g. Waugh and Hall (2002) and Kouznetsov et al. (2020)), …"
And line 66 has been extended to read:
"… We apply a correction for the non-linear growth of SF6 in the calculation of AoA, based on Fritsch et al. (2019), which allows a more quantitative view of the effect of the SF6 sinks."

*3. The conclusions are formulated quite vaguely. It should be clearly stated what are the findings of the present paper, and how they agree/disagree with earlier results, and what of the findings are new. It might make sense to have separate "Discussion/summary" (all references to earlier results etc.) and "Conclusions" (the concise statements that the authors are ready to defend).*

**Reply #3:**

Thank you for your comment. We extensively discuss the results in the various subsections of Section 3 and so we have decided on refraining from having a separate discussion section.
We have changed the Conclusion to state clearly that

- we show the effect of SF6 sinks on the AoA trend for the longer time series, not only for the short MIPAS time series which has large variability issues
- we rule out that other effects have an impact: we quantify these other effects and thus can say that the sinks do indeed have this big an effect on tracer derived AoA and its trends

For this, we have added to line 393:

"In this study we show the effect of SF6 sinks on the tracer derived AoA trend for both longer time series as well as for the short time series corresponding to the MIPAS time frame, whereby issues in variability are associated with the latter."

Line 401 has been amended to read as:

"Our sensitivity studies quantified that..."

Furthermore, as mentioned in Reply #2 and Reply #4, we have changed line 373 to highlight that EMAC is a chemistry climate model:

"… using the chemistry climate model EMAC…"

and line 374 has been amended to read:

"...This submodel allows for explicit calculation of SF6 sinks and we applied a correction for the non-linear growth of SF6 in the calculation of AoA (Fritsch et al., 2019)."

*4. Modelling studies of long-term evolution of SF6 distribution in the atmosphere have been reported by e.g. Reddmann et al (2001), Kovacs et al. (2017), Kouznetsov et al. (2020). The need for the present study and its similarities and differences from earlier ones should be clearly indicated.*

**Reply #4:**

Thank you for your comment. While the above mentioned modelling studies may bear some similarities, they also report a wide range of SF6 lifetimes, for example. The SF6 model used in this study is based on the reaction scheme of Reddmann et al. (2001).
We extended line 51 to read as:

"SF6 sinks lead to older apparent AoA (see, e.g. Waugh and Hall (2002) and Kouznetsov et al. (2020),..."

Line 56 has been extended to read:

"A more recent model study by Kovács et al. (2017), who used the Whole Atmosphere Community Climate Model (WACCM) to determine the atmospheric lifetime of SF6, reported a…"

Line 58 has been amended to:

"… Kouznetsov et al. (2020), who performed simulations of tracer transport with a chemical transport model, shows…"

Lines 372-374 have been changed to highlight the differences from earlier studies as well as clarify our aims:

"While previous studies (see, e.g. Kouznetsov et al. (2020)) showed that the chemical sinks can strongly influence SF6-derived AoA in terms of absolute values and decadal changes, we investigate for the first time how longer-term trends are affected in a consistent manner, and investigate the different contributions from circulation changes, changes in abundance of reaction partners, and trends induced by constant destruction rates. To make this step towards understanding the reasons for these discrepancies, we thus study …"

**Specific comments**

*Sec 2.1: A brief characteristic of the model is missing., e.g. "online spectral chemistry-climate model with hybrid sigma-pressure vertical layers".*

**Reply #5:**
Thank you for your comment. To meet your point, we extended the beginning of Sect. 2.1 (EMAC model), it now reads:
"For this study, we use the EMAC (ECHAM MESSy Atmospheric Chemistry, v2.54.0, Jöckel et al., 2010; Jöckel et al., 2016) model, a numerical Chemistry and Climate Model (CCM) system. It contains the General Circulation Model (GCM) ECHAM5 (ECMWF Hamburg, Roeckner et al. 2003), with its spectral dynamical core, as well as the MESSy (Modular Earth Submodel System, Jöckel et al. 2005; Jöckel et al., 2010) submodel coupling interface. The latter is a modular interface structure for the standardised control of process-based modules (submodels) and their interconnections. We apply the model in a T42 ...."

*Sec 2.2: The description of the SF6 sub-model is very unclear. Probably, most of the reactive species from Table 1 were not implemented as actual tracers in the model. One has to indicate which species were taken as climatology, which were forced from other models, and which were actual tracers. Was the submodel implemented as prescribed destruction rate as a function of altitude, latitude and season, or was it something more sophisticated? The description should be sufficiently detailed to allow for an independent reproduction of the experiment with another model.*

**Reply #6:**
Thank you for this comment, we improved the description of the SF6 submodel now such that it is clear what had been done and the simulations are reproducible. For this, we reduced Table 1 to only those chemical reactions that are actually considered in the module (following Reddmann et al., 2001). As the text already states, the submodel is indeed more sophisticated than prescribed destruction rates, it considers the individual chemical reactions for SF6 chemistry. As stated in Sect. 2.2, for the reactive species of SF6, tracers can be used in EMAC, but as noted in line 97-98, we here prescribe the tracer fields from an EMAC simulation with activated comprehensive interactive chemistry.
We additionally added in line 97-98:
"... The reactant species involved in the SF6 chemistry (HCl, H, N2, O2, O(3P) and O3) and the radiatively ..."
And moreover, we add to Sect. 2.3
"To compute the photodetachment rate of SF6-, we follow Reddmann et al. (2001) using the extraterrestial solar photon flux with no attenuation of the UV-photon flux, as provided by WMO (1986)."
To make clear that the submodel is part of the official MESSy distribution for all users, we further add in line 81:
"... in the mesosphere, and the submodel is operationally available for all users in MESSy from version 2.54.0 onwards."

*Contrary to stated in ll. 94-95, Fig.S1 does not show the relative importance of various reactions for SF6 destruction, but rather shows same reactions as in Table 1, but in a graphical form.*

**Reply #7:**
Thank you for your correction. We have amended the sentence in the revised manuscript to read as:
"... For a general overview of the various reactions see Fig. S1 in the Supplementary Information."

*Sec 2.3. This is by far not the first simulations of this kind. What are the similarities and differences from the setups used in earlier modelling studies? What is justification for specific model experiments, i.e. research questions to be addressed with each of the setups?*

**Reply #8:**
Thank you for your comment. To meet your point, we have added to the beginning of Section 2.3, starting at line 97:
"The simulations performed in this study include a more comprehensive approach for the calculation of the SF6 sinks. We use a climate chemistry model (as opposed to studies based on chemistry transport models, see e.g. Kouznetsov et al. (2020)) and use a more comprehensive SF6 submodel than previous chemistry climate model studies (see, e.g. Marsh et al. (2013) for the Whole Atmosphere Community Climate Model (WACCM))."
Regarding the CSS sensitivity simulation, line 108 now reads as:
"...simulation (year 1950 on repeat). With this simulation we aim to address the influence of the reactant species involved in the SF6 sink reactions. …"
For the time slice simulation TS2000, we added to line 111:
"This will allow us to investigate the effects of the SF6 sinks under a constant climate."

*Sec 3.1: The section has one comparison against 3-year-mean MIPAS profile for a latitude belt of 30N-50N, and one in-situ profile. Those are nice for illustrations, but are insufficient to judge on the model performance in reproducing SF6 distribution in the stratosphere.*

**Reply #9:**
Thank you for your comment. We agree that it would be desirable to carry out a more in-depth comparison to observations. However, we focus on the general effect of SF6 sinks on AoA, and so our aim here is to ensure a realistic-enough representation of SF6 by the EMAC model. Further studies would be required for better tuning of these, whereas we illustrate that the EMAC SF6 profile agrees well with SF6 profiles that are readily available - in this case, data provided by Ray et al. (2017) and Stiller et al. (2020).
We have added this in the beginning of Section 3.1, where line 167 now reads as:
"...observational data. This study does not perform a detailed comparison of SF6 profiles, as the major aim is not an in-depth evaluation of the SF6 submodel, but rather a quantification of the potential effects of the SF6 sinks on AoA and its long-term trends. However, to ensure that SF6 values in the EMAC model are within the range of observational estimates, we perform selected comparisons to data from Ray et al. (2017) and MIPAS SF6 (Stiller et al., 2020, paper in preparation. Fig 1a …"

*Fig.1a: It is not clear why this specific latitude belt, and these specific years were selected. The MIPAS error bars show standard deviation of individual MIPAS profiles. How those are related to the uncertainty of the average (of millions?) profiles that are shown? The MIPAS averaging kernel and spatio-temporal collocation notably affect the comparison (Kouznetsov et al. 2020). This issue has to be at least discussed.*

**Reply #10:**
Thank you for this comment. We are interested in the mid-latitude region and chose the specific years of 2007-2010 as there is a complete dataset in this time span. To make this clear, we add in line 170:
"… zonal mean SF6 averaged over 30°N-50°N and 2007-2010. These years have been chosen as the dataset is complete in this period."

With regards to your comment on MIPAS: In our approach we checked whether the mean model profile could be part of the ensemble of MIPAS measurements. We have not compared the mean profiles of the measurements and model. For our purpose, we need the standard deviation of the zonal mean ensemble. This is the error bar that is plotted in Fig. 1a. The standard deviation of the observed zonal mean ensemble consists of the measurement noise error of the MIPAS measurement, further random error sources from the retrieval (e.g. temperature uncertainties), the natural variability over the longitudes of the latitude band, and the four years of averaging (2007 - 2010). The model is not in contradiction to the observations if the standard deviations overlap. Averaging kernels and the error covariances both describe the interdependence of the vertical profile grid points in the case of the measurements. Such an interdependence tends to smooth the profiles in the retrieval application chosen for MIPAS SF6. Since we expect rather smooth profiles anyhow, partly because of the nature of this trace gas, and partly due to averaging over a large number of profiles, further smoothing does not greatly change the resulting standard deviation. If we wanted to compare the mean profiles to each other, the correct approach would be to apply the averaging kernel of each single MIPAS profile to the model profile at the geolocation of the MIPAS measurement, and to average over the smoothed model profiles. This, however, was not our intention.

We now include this in the discussion of Fig. 1a, in line 170:

"The error bars represent the standard deviation of the zonal mean ensemble, which consists of the measurement noise error of the MIPAS measurement, further random error sources from the retrieval (e.g. temperature uncertainties), the natural variability over the longitudes of the latitude band, and the four years of averaging (2007 - 2010)."

*Fig 2a. Interestingly, Kouznetsov et al. (2020, Fig 5 there) shows very similar offset of the model profiles with respect to the in-situ one. I wonder if it is a coincidence, or an indication of a similar issue with both model setups.*

**Reply #11:**

Indeed, this is an interesting point! As there are many differences between the model types and setups, and also considerable uncertainty in the satellite derived SF6, it is hard to judge whether the similarity might stem from a similar issue.

*Sec 3.2: The methodology for the lifetime estimate is quite unclear. Instead of explaining the method used, the authors put a reference to a 600-page report (Braesicke et al, 2019). The method, probably, refers to the equation in p. 1.20 of the report. The equation assumes well-mixed atmosphere and implies that the destruction of SF6 is proportional to its burden, which is not the case: the destruction does not depend on the tropospheric content of SF6, but rather on its content in mesosphere. In a situation when the change of SF6 burden is substantial at a time scale of ~10 years (AoA in the stratosphere) the difference leads to "surprising" results like reduction of SF6 lifetimes by 25% over 100 years. Given the slow destruction of SF6 one could still define the lifetime in terms of well-mixed assumption, but that would require a long-term simulation without emissions, to let the mixing ratio relax to its equilibrium distribution and get to the exponential decline of the total burden. Alternatively, as it was done by Kouznetsov et al. (2020), one could use a total burden that corresponds to the mixing ratio next to depletion layers.*

**Reply #12:**

Thank you for this comment, we improved the description of the SF6 lifetime calculation such that it is clear what has been used. To make this clear, we add in line 198:

"… We calculate the lifetime following Equation 1 in Section 3 of Reddmann et al. (2001), namely: [see Manuscript]. It is based on the reaction rates (ki) of the chemical reactions (Ri) marked in Table

1, the branching fraction e (taken as 0.999, see Reddmann et al. (2001)), and the efficiency of the SF6-recovery reactions (n), where n is calculated as: [see Manuscript]."

*Sec 3.4 -- 3.5: Same behaviour of trends in the apparent SF6 AoA has been pointed out by Waugh and Hall (2002), Waugh et al.(2003) and demonstrated with extensive model simulations (Kouznetsov et al., 2020) for various latitudes and altitudes. Please specify what is a new finding here with respect to those studies.*

**Reply #13:**
Thank you for your comment. As addressed in Replies #2, #4, #8. Kouznetsov et al. (2020) uses a chemisty transport model and considers the AoA trend only over 10 years, whereas we look at long-term trends and use a general circulation model with atmospheric chemistry. While our results bear similarities to the Whole Atmosphere Community Climate Model (WACCM) , see e.g. Marsh et al. 2013, they do not use a precise correction for the non-linearity of SF6 emissions and are therefore not in line with Engel et al. (2017) nor with Fritsch et al. (2019). We have, for example, amended the Conclusion (see Reply #4) and Section 2.3 (see Reply #8) to clearly state these differences.

*l.329-365: The fact that SF6 destruction causes the positive trend in the apparent AoA follows from the simple fact that the SF6-AoA is proportional to a difference between stratospheric and tropospheric SF6 mixing ratios. Since the destruction is proportional to the SF6 mixing ratio, the difference increases together with the increase of the atmospheric SF6 burden. There is no need to involve any equations or advanced concepts (like Green functions etc.) to explain that.*

**Reply #14:**
Thank you for your comment. We find the additional mathematical presentation of the effect of SF6 sinks on AoA could be of interest for the reader. Furthermore, as discussed with Referee #1 (see Main Comment #2), we extended Section 3.5 to now include an examination of the possibilities to include a correction for the SF6 sinks.

---

## Author Response (AR2)

**Author's Response to Report #1**

1. The main objective(s) of the study are still not clearly declared. A phrase has been added as a response to my first comment "... with the aim to understand the effects of SF6 sinks on tracer derived AoA and its long-term trends. Specifically, we calculate for the first time the effect of the sinks on the long-term trend of AoA."

This phrase is by no means sufficient. The latter sentence should have and "apparent AoA" (or "SF6 apparent AoA") rather than AoA. SF6 sinks, probably, have a minor effect on AoA via radiative forcing and circulations, but it is well beyond the accuracies of current models.

I would be expecting formulation of the objectives in a way they can be clearly shown as achieved in the conclusions, and streamlining of the whole narrative to meet the objectives.

**Response.** The main objectives of the study are stated in the last paragraph of the Introduction, and we extended the sentences to emphasize the specifics of the study even more. In the newest version of the manuscript:

- Line 70 ff. "[...] we apply the chemistry climate model [...] with the aim to understand the effects of SF6 sinks on tracer-derived AoA and its long-term trends."
- Line 72 ff. "Specifically, we calculate for the first time the effect of the sinks on the long-term trend of SF6-derived AoA, and quantify how this effect is modulated by circulation changes, or by changes in the abundance of relevant species for SF6 chemistry."
- Line 79 ff. "We apply a correction for the non-linear growth of SF6 in the calculation of AoA, based on Fritsch et al. (2019), which allows for the quantification of the effect of SF6 sinks on SF6-based AoA in isolation.

We re-state our objectives in the Conclusion:

• Lines 481- ff. "We investigate for the first time how longer-term trends are affected in a consistent manner and investigate the different contributions from circulation changes, changes in abundance of reaction partners, and trends induced by constant destruction rates."

And state the corresponding achievements in the paragraphs that follow:

- Refer to Lines 500-505 ff for result on quantification of AoA climatology.
- Refer to Lines 506-517 ff for results on quantification on AoA trends.
- Lines 518 ff. "Our sensitivity studies quantified that the positive AoA trends are neither a result of climate change, nor of changes in the substances involved in SF6 depletion. The SF6 sinks themselves are the reason for the increase in apparent AoA."
- Lines 523 ff. "[...] we can conclude that SF6 sinks have the potential to explain the longlasting AoA trend discrepancies between models and observations."

We continue in the Conclusion with the final objective, namely:

- Line 524 ff. "[...] we put forward a first approach towards a method for SF6 loss correction"
- Lines 525 ff. "[...] we can conclude that a linear correction [...] can likely be applied to AoA values up to 4 years."

To ensure clarity for the reader, the aforementioned objectives have been clarified in the Introduction. This paragraph now reads as follows (starting on Line 70 ff in the newest version), with the new additions highlighted in bold:

"In the present study, we apply the chemistry climate model EMAC (ECHAM MESSy Atmospheric Chemistry, Jöckel et al., 2010; Jöckel et al., 2016) with the aim to understand the effects of SF6 sinks on tracer-derived AoA and its long-term trends. Specifically, we calculate for the first time the effect of the sinks on the long-term trend of **SF6-derived AoA**, and quantify how this effect is modulated by circulation changes (recent climate change), specified model dynamics, or by changes in the abundance of relevant species for SF6 chemistry. Furthermore, we analyse the contribution of the SF6 sinks themselves on the long-term trend of SF6-based AoA. As an outlook, we thereupon provide first thoughts on how to apply an AoA correction to observations taking SF6 sinks into account. The chemistry climate model uses the second version of the Modular Earth Submodel System (MESSy2) to link multi-institutional computer codes. In our simulations, we employed the MESSy submodul "SF6" which explicitly calculates SF6 sinks based on physical processes (based on Reddmann et al., 2001), rather than on crude parameterisations. We apply a correction for the nonlinear growth of SF6 in the calculation of AoA, based on Fritsch et al. (2019), which allows for the quantification of the effect of SF6 sinks on SF6-based AoA in isolation. In Sect. 2 we describe the EMAC model and the SF6 submodel as well as the observational data we use for comparison. Sect. 3 contains a comparison of the EMAC climatologies with MIPAS data, a comparison of the EMAC trends with MIPAS and balloon borne measurements and an analysis of the results of two sensitivity simulations. The model results are discussed in the following using theoretical considerations of the effects of sinks on AoA trends (Sect. 4), including first thoughts on possible correction methods for the sinks, that are highly desirable for the use of observational data. In Sect. 5, we discuss the results and provide some concluding remarks."

The way SF6 sinks affect apparent AoA and its trends have been shown earlier in several studies (though with theoretical considerations and somewhat shorter simulations), and the conclusions made are quite unequivocal: "The apparent over-ageing introduced by the sink is large and variable in space and time. Moreover, the over-ageing due to the sink increases as the atmospheric burden of SF6 grows."

The need for longer-time simulation has to be justified, and research questions to be addressed with the simulations have to be formulated.

**Response.** It is correct that the above quote from Kouznetsov et al. (2020) shows the effect of SF6 sinks on apparent AoA. However, their calculation of the apparent AoA trend runs over a short time period of only 11 years. There has been a long-standing discrepancy between observations and model simulations of AoA, and in particular, of the AoA trend over several decades. The study of Engel et al. (2009) presents a multi-decadal time series of observational data and analyses the trend of observation-based AoA spanning 30 years. While it has been previously shown that SF6 sinks lead to an over-aging of apparent AoA, the long-term effect of it has not yet been investigated. We thus investigate the long-term effects of the SF6 sinks and present these in relation to the observational data spanning several decades in Engel et al. (2009). These explanations have been detailed in the introduction of the paper. Moreover, our study quantifies the modulation of the long-term SF6 effect on the AoA calculation by changes in the reactant species, circulation changes and different representations of large-scale dynamics (via specified dynamics; see above), and as such, goes clearly beyond the mere conclusion that the over-ageing due to the sinks increases with growing SF6 concentrations.

2. The new version of the manuscript states that "a comprehensive understanding of what contribution the individual effects have on the AoA trend depending on altitude and latitude is still missing". A comprehensive understanding is a subjective and poorly defined matter, and thus does not qualify for an objective. One could put as an objective to check if the conclusions of earlier studies are reproducible with longer simulations and more sophisticated model. A valid aim would be to quantify influence of the individual effects on the apparent AoA trend, or evaluate their relative importance. However, explicit list of the individual effects that are under consideration is mandatory in this case. The paragraph above the phrase lists them: acceleration of the BDC, concave growth rate of tropospheric SF6, sparse sampling of in-situ observations, differences in the changes between the deep and the shallow BDC branch, integrated effect of mixing, SF6 sink. However, of all those, the paper considers only SF6 sink (WS vs NS cases, CSS case) and acceleration of the BDC (REF vs TS2000). The correction for the concave growth rate of tropospheric SF6 has been just applied, and the parameters of the correction were chosen to get the ages from (SF6,NS) case matching those from

(lin,NS), essentially making one of these tracers redundant, and leaving the effect of concave growth rate beyond the consideration. The effect of the changes in the reactant species has been considered in the paper, but not listed here. Besides those, a gravitational separation (Kovacs 2017, Kouznetsov 2020) has been considered earlier and shown to have an effect, though much smaller than one of the mesospheric sink.

**Response.** The sentence referred to by the Reviewer concludes a paragraph on the different processes that influence (SF6-derived) AoA trends, and it is not claimed here that this sentence summarizes our objectives. Rather, the aim of our study is stated in the last paragraph of the Introduction (Line 70- ff). Clearly, we are not claiming that our paper will achieve a comprehensive understanding of the entire topic, only that it will contribute another puzzle piece for the overall picture. In this paper we are particularly seeking to quantify the effects of the SF6 sinks on SF6-derived AoA climatologies and (long-term) trends, thereby analysing the modulation of those effects through circulation changes or changes in reactant species (see our answer to point 1). Moreover, with our developments, we pave the way for more in-depth studies that can tackle the other issues the Reviewer lists and are also listed in our introduction.

Further, we strongly disagree with the critical comment that through the correction for the nonlinear (concave) growth of SF6 we make the non-linear passive SF6 tracer ("(SF6,NS) ") redundant and " leave the effect of concave growth rate beyond the consideration" – indeed this correction is a prerequisite to study the effects of SF6 sinks on SF6-derived AoA.

An ideal Age tracer should satisfy two conditions: a) growing linearly over time, b) being passive. When deriving AoA from SF6, both assumptions are violated. Thus, the effects of SF6 sinks on SF6derived AoA can only be quantified in isolation, if the first violated assumption is taken care off. In an earlier publication by our group, we extensively studied and tested the methods to correct for the non-linear growth in detail (Fritsch et al., 2020), building on the exact same method that is used in many observational studies (Engel et al. (2009)). This implementation of the correction for non-linear growth is the prerequisite for the exclusive analysis of the effect of the SF6 sinks, thereby extending our work beyond other recent work.

3. The conclusions now admit that a major effect of sinks on the apparent SF6 AoA and its trends was shown earlier. The minor missing part now is the statement that the current study confirms/disproves those findings.

**Response.** What was shown earlier in Kouznetsov et al. (2020) is that SF6 sinks have an effect on apparent AoA and its trend calculated over 11 years. For this, a CTM was used. What was not discussed in the previous study is what effect SF6 sinks have on **long-term** trends (i.e. over several decades) of apparent AoA. It is precisely the latter effect that we investigate in this study using a GCM. We have amended the sentence starting in Line 480 ff to read as: "In agreement with our results, previous studies (see e.g. Kouznetsov et al. 2020) showed that the chemical sinks strongly influence SF6-derived AoA in terms of absolute values and decadal changes. We investigate the ..."

4. The new text states "previous studies ... showed that the chemical sinks can strongly influence SF6derived AoA in terms of absolute values and decadal changes". Indeed it has been earlier shown that the chemical sinks do strongly influence SF6-derived AoA terms of absolute values and trends, not not just that they can.

**Response.** The 'can' has now been removed.**

5. The only major change in the manuscript since the previous revision is re-writing and changing of the scope for the section 4. The section is now called "Theoretical considerations and concept for sink correction methods", and formulates a correction procedure for over-aging that (as Conclusions state) "can likely be applied to AoA values up to 4 years".

The section is weakly linked to the rest of the paper. The formulations there rely on obviously violated unrealistic assumptions (delta-function age spectrum and existence of the "effective lifetime" as a function of true AoA), the experimental evidence provided only for a linearly-growing SF6 and only for 30-50N latitude belt. The resulting conclusions are vague.

The feasibility of such a correction has been challenged (Kouznetsov 2020, see also notes about the 1D-model behind the correction for non-linear growth by Waugh and Hall, 2002). If one decides to propose such a correction, it deserves a separate paper with a consistent formulations and strong evidence. The applicability limits, uncertainties, and error-propagation through the correction have to be carefully considered there.

Therefore I repeat the request from the previous-stage review to discard this section from the present paper.

**Response.** The addition of the concept for a sink correction method to Section 4 was included in response to a comment in the first review by Eric Ray, who stated that *"this paper is an important step forward in our understanding of mean ages derived from observations and how models can be used to help put them in context. My main comments described below revolve around how best to use this information to help us make the observationally derived SF6 mean ages more accurate."* Further, he commented: *"As a corollary to your findings, couldn't a correction to SF6 mixing ratios be made in the calculation of mean age to account for loss?"*.

This prompted us to explore first concepts of how, and under which circumstances, such a correction could be applied. We linked the theoretical considerations to the data we obtained from our simulations. In the theoretical considerations, we state that under the - hypothetical - assumptions of a delta-function age spectrum and an "effective lifetime" that is independent of AoA, we would have a linear relation of the ideal mean AoA and the SF6-derived mean AoA. As stated by the Reviewer, those assumptions are obviously unrealistic. However, with this thought experiment, we can see under which assumptions we would have a strictly linear relation, and thereafter test how strongly the violation of the assumptions lead to deviations of the linear relation. We now clarified that the assumptions are rather hypothetical in the text:

Line 427 ff: "As a thought experiment, and with the aim to derive an analytical concept for the correction of mean AoA for the sinks, we make the hypothetical assumption that the age spectrum is represented by a single, average path, [...]."

Line 439 ff: "However, if the effective lifetime were constant, the apparent mean age  $\tilde{\ell}$  would be linearly related to the actual mean AoA."

Line 445 ff: "While the assumptions will clearly be violated in the model, we investigate, based on data from our model simulations, whether the violations might effectively be small enough so that the linear relation still holds."

We fully agree with the Reviewer in that this topic does deserve a separate paper, in which the applicability limits (for example with respect to the non-linear increasing tracer, and different regions of the atmosphere) are explored. This had already been clearly stated in the last paragraph of this section, where we also propose a way forward to constrain the SF6 sinks from observational data. However, we are convinced that those first considerations explored here will be useful for a much needed follow up study, and might indeed prompt work in this direction.

The last paragraph (see Lines 464-475 ff) has been amended to make the above points clearer:

"Overall, the results derived in this section indicate that a correction of observational SF6-derived mean AoA for the effects of chemical sinks is likely possible by applying a time-dependent linear correction function. This linear relation between AoA from the ideal and from the chemically depleted SF6 tracer holds for mean ages below about 4 years. Here, we show this relation for the linearly increasing tracer in northern mid-latitudes. However, further analysis indicates that this linear relation also holds for the realistic SF6 tracer with non-linear growth over time (not shown). Furthermore, the linear relation seems to be nearly identical for different latitude bands (not shown), which is a very promising property for future applications of a correction method.

We emphasize that the strength of the SF6 sink in our model simulations is not well enough constrained to properly establish such a correction function. Deriving suitable values for the linear relation between  $\Gamma$  and  $\tilde{\Gamma}$  (and thus the effective lifetime  $\tau$ eff) should be obtained by means of observational data. This could be achieved by using simultaneous measurements of SF6 and other age tracers, as previously shown by Leedham Elvidge et al. (2018) and Adcock et al. (2021). Furthermore, the concept needs to be evaluated vigorously on model data to assess its errors and limitations."

**Further comments:**

Sec 2.2: Now the implemented depletion mechanism is more clear. If I got it right, the model dynamics in the present study is not affected by any of the simulated tracers. Moreover, the reactant species and photon flux are prescribed and are not coupled to the rest of the model dynamics. This essentially means prescribed loss rate as a function of time and location. Is there any principal difference between an online CCM and an offline CTM driven by a climate model in this case?

**Response.** You would be correct if we only considered the nudged simulation. The SF6 sinks are not fully interactive, but this would be a negligible effect. However free-running simulations, future projection simulations, and time-slice simulations can only be run in a general circulation model. These experiments are not possible in a CTM.

For our purposes it was enough to have all the species prescribed. With our CCM we could, in principle, run the species interactively. This work sets the basis for further studies in this direction, which would not be possible with a CTM.

Sec 2.3: The SF6 emissions (prescribed lower BC for SF6 concentrations) should be described here rather than in sec 2.5. Does "linear emission" mean linear growth of prescribed SF6 mixing ratio at the lower BC? I would note that it does not correspond to a constant or linearly-growing emission rate, which one might think of when reading "linear emission". How exactly the lower BC was prescribed? Was it just a uniform mixing ratio over the surface as a given function of time? A reference to the origin of the data for that function would be needed. How the lower BC were specified for the PROJ run?

**Response.** The SF6 emissions you are referring to are the surface boundary conditions prescribed as mixing ratios. We state clearly in Line 181 ff of the newest version: "Technically, these 'emissions' are realized via lower boundary conditions in our simulations". For clarity we amended the sentence to: *"Technically, in our simulations these "emissions" are realised via lower boundary conditions, which are based on surface observations, as in Joeckel et al. (2016)".*

As these boundary conditions come hand in hand with the tracers we use in our analysis, we feel that this information belongs to Section 2.5: Analysis method.

Fig.1a. (Not fully addressed in reply #10 from previous review) Completeness of the dataset is a valid criterion. The request to justify the use of a specific latitude belt (comment 12 from previous review) has not been addressed.

**Response.** We choose the mid-latitude region as we compare our results with the timeseries between 30-50N provided by Engel et al. (2009), see Fig. 4. Adding to this, as pointed out in the paper, we do not focus on an in-depth comparison of SF6 profiles. The text clearly states this in Lines 195-198 ff: "This study does not perform a detailed comparison of SF6 profiles, as the major aim is not an in-depth comparison of the SF6 submodel, but rather a quantification of the potential effects of the SF6 sinks on AoA and its long-term trends. However, to ensure that SF6 values in EMAC are within the range of observational estimates, we perform selected comparisons to data from..."

The contribution of the noise error into the error of zonal-mean profile averaged over several years is negligible, contrary to errors arising from averaging kernel and systematic measurement errors. Looks like the terms "standard deviation of profiles over the averaging range" and "standard error of the mean profile" are confused. The latter is the right measure of the mean-profile uncertainty which should appear as the error bar.

**Response.** The vertical resolution for the MIPAS SF6 profiles is relatively coarse, however, since the zonal mean profiles are rather smooth and do not show pronounced vertical structure, the impact of the limited vertical resolution can be neglected. Regarding the sampling, it is correct that the average profile differs for a sampling on a regular model grid versus sampling on MIPAS geolocations; however, our aim is to demonstrate that the model profiles come closer to the MIPAS profiles by consideration of a more realistic source evolution and the accounting of the mesospheric sink, and a quantitative agreement is not our primary aim (this would require knowledge about the ionization state of the mesosphere along the SF6 trajectory, which we do not have).

We have amended Figure 1a. to show both the systematic error as well as the standard deviation due to month-to-month variability. Thus, the standard deviation is now comparable between MIPAS and the EMAC model data, both depicting the standard deviation of monthly mean, zonal mean SF6 averaged over 30N-50N for the time period 2007-2010. As such, the caption of Figure 1 has been changed to include the following:

"Black error bars depict the standard deviation of MIPAS SF6, pink error bars the systematic error of MIPAS. The systematic error comprises of errors in the spectroscopic data and uncertainty in the instrumental lineshape, which results in a systematic error of 2% for the lower (10 km) and 11% for the upper (60 km) stratosphere. See Stiller et al. (2020) for details."

**Author's Response to Report #2**

We thank the Reviewer for his positive and constructive assessment of our work. Our changes to both text and figures in response to the Reviewer's comments are summarized below. These changes include modifying Figure 5 to provide the Reader with a more intuitive and clearer picture to convey the sink-impacts versus variability of the trend from SF6-derived AoA. Furthermore, we corrected an error in the calculation of the variability of the trends in Table 3.

Reply to #1 and Figure 3: The addition of Figures 3g and h are great. I don't necessarily agree with your interpretation of Figure 3h though. The first issue is I actually don't see the text you have in the author response in the revised paper. It's not at line 229 and I couldn't find it later in the paper. Assuming that text does go in the paper, the differences in the early period look to be negligible, less than 0.2 years, for ages less than 5.5 years and although the differences are larger in the later period, they still appear to be quite small for ages less than 5 years. You mention differences greater than 20% for ages greater than 3 years but I don't see that. There would have to be differences of 0.6 years or more for this to be true and I don't see those red contour values except in the polar regions at ages older than 5 years. So something isn't consistent with the figure and your description of it. Also, since you talk about the figure in percentage differences then you should really plot those rather than absolute differences to make it easier for the reader to follow.

**Response.** First, we apologize for referring to the wrong line number in our previous reply. The text we added in the previously revised paper was at line 261. Now, in the newest version, the text is at line 268 ff.

Furthermore, the deviating interpretation of Fig. 3g and h appear to be a result of a misunderstanding, caused by an erronous labeling of the colorbar in the previous version of the figure. Indeed, Figures 3g and h show the relative difference between SF6-derived AoA with sinks and without sinks for the two time periods. We have corrected the labels in Figures 3g and h and have amended the caption to include the equation used to calculate the relative difference in SF6-based AoA. We hope that with this clarification the misunderstanding is removed, and the Reviewer agrees with our interpretation.

Figure 5 caption: This figure is a bit confusing. The use of both dotted lines for negative trend values and dotted regions for not significant trends really threw me for a while. Is the only dotted region at the bottom of Figure 5b? Also, what are the units of the shaded contours in Figure 5? Percent? It should be labeled. Is this AoA(WS,SF6) - AoA(NS,SF6)? Because if so the values should be positive for both plots shouldn't they? How can the differences in the trends flip sign from trends over the whole period to trends over the later period? Looking at Table 3, the trend differences at that location should be 0.25 years/decade for the whole period and 0.29 years/decade for the later period.

Lines 334-339: This added paragraph is not clear to me, maybe because I'm not clear about Figure 5. The phrase 'deviates beyond 100% (i.e. by a reversal of sign)' should be rephrased. If the actual AoA trend were small and negative and the trend with sinks was small and positive, they could still be in good agreement within uncertainties. The sign reversal does not automatically make the agreement poor. When I look at the trend numbers in Table 3 it appears that the AoA(WS,SF6) trend from 1965-2011 and 30-50N and 30 km is 0.19+/-0.26 and for AoA(NS,SF6) it is -0.06+/-0.09. The value of AoA(NS,SF6) should be what is plotted in Figure 5a but it appears to be -0.006 in the plot. Is the plotted value in years/year rather than years/decade? If so, that should be changed so the numbers agree between the figure and the table. Then looking at the uncertainties on these trends they appear to be in agreement within the uncertainties. The uncertainty range on both trends span positive and negative values.

**Response.** Again, we apologize for unclear labeling of the figure. Indeed, the figure in the previous version did show the relative deviation. The motivation for this figure was to find out up to which altitude the SF6-based AoA trends can be trusted without considering sinks. For this, we examined from which region (i.e. altitude) upwards the trends are more strongly influenced by the SF6 sinks than by uncertainty due to variability. To answer this, we initially (in the previously revised paper) showed the relative trend differences between REF(WS, SF6) and REF(NS, SF6) for different time periods in Figure 5. As we now understand that this figure was not very clear for the reasons stated by the Reviewer, we changed it to the vertical profile of the AoA Trends (30°-50°N). This new version of the figure more directly conveys the sink-effects versus variability of the trend for different time periods. In other words, Figure 5 now shows profiles of the absolute trend and its uncertainty (2 $\sigma$ ) of SF6-derived AoA (with and without sinks) over 30°-50°N for different time periods, so that it can be directly extracted from the figure for which altitude range the trends differ significantly. Therewith, the figure provides additional information on the variability of the trend for the 10-50 km altitude range, as opposed to only at 30km in Table 3.

This figure indicates that the trend for the period from 1965-2011 is largely unaffected by the SF6 sinks until about 20 km altitude. For the trend in the shorter time period of 2000-2011, effects from the SF6 sinks become visible at about 22 km altitude and higher, which is mainly due to the larger uncertainty stemming from a shorter time period. This means that in the mid-latitudes, the SF6-based AoA trends with and without sinks are not distinguishable from each other up to 20 km and 22 km altitude, depending on the period.

The uncertainty in the trend calculated from SF6-based AoA with sinks increases with increasing altitude – this is to be expected, as the effect of the SF6 sinks increases with increasing altitude. Furthermore, the trend of AoA(NS, SF6) over the years 2000-2011 is significantly positive at 25 km and higher, as opposed to a negative trend over the longer period 1965-2011. This is the reason for the reversed sign in the previous version of the figure, which was indeed confusing. This trend calculation is based on a very short time period of approx. 1 decade, so that inter-annual variability strongly influences the trends (see e.g. Dietmüller et al., 2021). This also causes the apparent discrepancy between the positive trends shown in the new Figure 5 (which were calculated with a simple linear fit) and the trends listed in Table 3 for a similar time period. The latter were calculated with A regression model taking other variability modes into account to enable a comparison with MIPAS.

The text in the manuscript has been updated to include these details. Now, in the newest version, the text to Figure 5 can be seen from line 337 ff. onwards.

Line 430: switch 'two latter' to 'latter two' **Response.** This has been corrected.